# Thalamocortical control of cell-type specificity drives circuits for processing whisker-related information in mouse barrel cortex

Timothy R. Young [1], Mariko Yamamoto [2], Satomi S. Kikuchi[1], Aya C. Yoshida[1], Takaya Abe [3], Kenichi Inoue[3], Joshua P. Johansen [4], Andrea Benucci[5,6], Yumiko Yoshimura[2] & Tomomi Shimogori[1] ✉

Excitatory spiny stellate neurons are prominently featured in the cortical circuits of sensory modalities that provide high salience and high acuity representations of the environment. These specialized neurons are considered developmentally linked to bottom-up inputs from the thalamus, however, the molecular mechanisms underlying their diversification and function are unknown. Here, we investigated this in mouse somatosensory cortex, where spiny stellate neurons and pyramidal neurons have distinct roles in processing whisker-evoked signals. Utilizing spatial transcriptomics, we identified reciprocal patterns of gene expression which correlated with these cell-types and were linked to innervation by specific thalamic inputs during development. Genetic manipulation that prevents the acquisition of spiny stellate fate highlighted an important role for these neurons in processing distinct whisker signals within functional cortical columns, and as a key driver in the formation of specific whisker-related circuits in the cortex.

The mammalian neocortex displays considerable diversification among its cell types and connections, underlying the complexity of neural circuits involved in processing information and shaping emergent properties within different brain regions. The compartmentalization of the rodent somatosensory cortex into distinct whisker-related circuits (referred to collectively as the barrel cortex) is an example of such diversity[1]. Here, the unique cytoarchitectural arrangement of cortical layer 4 (L4) neurons into whisker-specific "barrels" and intervening "septa" regions forms the basis for conceptualizing the barrel cortex into two functional divisions that process and integrate distinct streams of whisker-evoked signals[2–6].

Mirroring the topographic, grid-like organization of whiskers on the contralateral whisker pad, L4 "barrel" regions (sometimes referred to as "hollows" to distinguish them from barrel cortex itself) are highly populated by clusters of spiny stellate neurons, with asymmetric dendritic fields (lacking an apical dendrite) directed toward dense thalamocortical axon (TCA) terminals that represent input arising from a single whisker. Located in the narrow regions interspersed between each barrel, L4 "septal" neurons possess a pyramidal morphology with distinct connectivity compared with their neighboring neurons in the barrel[7]. Whereas L4 stellate neurons receive thalamic input from ventroposteromedial (VPM) nucleus and have local connectivities that are mainly confined to within their home barrel column, L4 septal neurons display more widespread cortical interactions with their inputs coming from the medial part of the posterior medial (POm) thalamic nucleus[8–10]. As such, barrel and septal circuits broadly refer to functional cortical circuits that are vertically aligned and connected with their corresponding L4 cell types/regions, distinctly encoding for

[1]Laboratory for Molecular Mechanisms of Brain Development, RIKEN Center for Brain Science, 2-1 Hirosawa, Wako, Saitama 351-0198, Japan. [2]Division of Visual Information Processing, National Institute for Physiological Sciences, National Institutes of Natural Sciences, Okazaki 444-8585, Japan. [3]Laboratory for Animal Resources and Genetic Engineering, RIKEN Center for Biosystems Dynamics Research, 2-2-3 Minatojima-minamimachi, Chuo-ku, Kobe, Hyogo 6500047, Japan. [4]Laboratory for Neural Circuitry of Learning and Memory, RIKEN Center for Brain Science, 2-1 Hirosawa, Wako, Saitama 351-0198, Japan. [5]Laboratory for Neural Circuits and Behavior, RIKEN Center for Brain Science, 2-1 Hirosawa, Wako, Saitama 351-0198, Japan. [6]School of Biological and Behavioural Sciences, Queen Mary University of London, London E1 4NS, UK. ✉e-mail: tomomi.shimogori@riken.jp

whisker-specific spatiotemporal information and multi-whisker kinetic information, respectively[7,8,11].

To date, the mechanisms underlying the initial formation of barrel cortical circuits are not completely understood[12,13]. Given that it represents a major input layer for receiving distinct whisker information streams, L4 barrel cortex likely serves as a platform through which formation of barrel and septal circuits emerges. Beyond the intrinsic patterning mechanisms of the cortex[14] and before the peak of cortical innervation has occurred, the fate of postmitotic L4 neurons at birth is yet to be fully committed[15,16]. Recent evidence suggests that L4 diversity may arise in a TCA-dependent manner, with spiny stellate morphology being sculpted from a common pyramidal-like progenitor[17,18]. Manipulation of TCA input to the cortex has resulted in disruption to overall barrel cytoarchitecture and broad changes in cortical gene expression[19–21], while several studies have highlighted genetic factors (both cortex- and thalamus-specific) that are involved in proper barrel formation[22–25]. Here, however, a detailed molecular characterization underlying the diversity of L4 barrel hollow and septal neurons has not been fully addressed. Moreover, despite well-characterized cytoarchitectural and connectivity differences, L4 barrel cortical cell types cannot be easily separated on the basis of their electrophysiological properties[26]. Thus, a closer genetic dissection of barrel/septa circuits and their formation (for example, in the case of TCA ablation and knockout models) is required.

In order to investigate this, we therefore aimed to identify genetic markers that can be used to distinguish between L4 domains within mouse barrel cortex. Utilizing a genome-wide spatial transcriptomics platform, we identified reciprocal barrel hollow- and septa-related patterns of gene expression that displayed a close spatiotemporal relationship with thalamic innervation of the cortex and the acquisition of cell morphology. Further, we found evidence of a default cortical state that is of septa/pyramidal identity, upon which specific TCA input is required to further pattern the primary somatosensory cortex into its barrel and septa divisions. Ectopic expression of a septa-related gene, *Smad7*, prevented L4 neurons from acquiring spiny stellate morphology, even with TCA innervation intact. Loss of stellate-populated circuits resulted in an increase of septa-like connectivity, more widespread whisker-evoked cortical activation, and an impaired ability to perform whisker-dependent texture discrimination.

Overall, these experiments highlight the molecular diversity in L4 barrel cortex that may underlie the development and/or function of barrel and septal circuits. Further, they establish an important role for specific TCA-derived cues (activity and/or molecular) in regulating L4 cellular diversity, highlighting a key driving influence on the formation of specific whisker-related circuits in the barrel cortex.

## Results

### Utilization of spatial transcriptomics to aid discovery of whisker-related gene expression

There are currently few reported genetic markers which demonstrate specific expression relating to neurons in barrel compartments[27–30]. To aid the further discovery of such genes in mouse barrel cortex (Fig. 1), we utilized Dlx5/6[IRES-Cre]; Celsr3[flox/flox] (Celsr3 conditional knockout) mice which lack TCA innervation of the cortex and as a result, fail to form barrels despite normal cortical lamination and positioning of L4 neurons over the period that the barrels normally form[31] (Fig. 1b and Supplementary Fig. 1a–c). With sparse fluorescent protein (FP) labeling of L4 neurons using in utero electroporation (IUE), we further noted a lack of neurons with spiny stellate morphology (11% vs 58% in controls) and a corresponding increase in pyramidal-like neurons (89% vs 42% in controls; $\beta_{genotype} = 2.41$; s.e.=0.37; $\sigma_{animalID} = 0.38$; generalized linear mixed-effects model with parametric bootstrap test, $p = 0.0002$) within the S1 cortex region of these mice (Fig. 1c, d). As has been reported for L5, these L4 pyramidal-like neurons also displayed signs of abnormal dendritic development (Supplementary Fig. 1c), which

may arise from the lack of extracortical inputs supporting their normal growth and formation[31].

We hypothesized that such alterations to the cell-types in L4 "barrel" cortex would also be associated with changes in gene expression relating to the barrel and septa compartments. Therefore, using the 10x Visium platform, we generated and compared spatial profiles of gene expression in primary somatosensory cortex (S1) of control and Celsr3 cKOs (Fig. 1e–g). To narrow down our analyses to L4 barrel cortex, we employed a publicly available annotated scRNA-seq reference dataset of postnatal day (P) 7 somatosensory cortex to predict the locations of "L4 spots" within each profiled section, implemented with a deep-learning-based integration method[32] (see Methods). Differential expression analysis was performed using RNA counts from each L4 spot (control: $n = 73$; Celsr3 cKO: $n = 85$ spots; 3 sections per genotype) with candidate barrel hollow- and septa-related genes being downregulated (577 genes) and upregulated (81 genes), respectively, in Celsr3 cKOs. To overcome possible technical limitations from fluctuations or dropout events leading to false negative gene counts in our spatial data[33,34], we further utilized Tangram spatial/single-cell integration to predict scRNA transcript distribution across control sections. Candidates were initially screened for their barrel cortex expression patterns using this scRNA-integrated spatial transcriptomics data, followed by in situ hybridization (ISH, Fig. 1h, i and Supplementary Fig. 1d–g).

While the Visium platform was insufficient to resolve gene expression between distinct barrel compartments, we hypothesized that individual L4 spots (Ø = 55 μm; center-to-center spacing = 100 μm), depending on its tissue placement, would each sample distinct proportions of barrel and septa neurons, and thus be reflected in their gene counts. Therefore, to expand and refine our candidate list, semi-supervised clustering of L4 scRNA-mapped spatial data ($n = 184$ spots, 11 sections) was performed (Supplementary Fig. 2a–d), guided by our screened list of candidates ($n = 38$ genes, see Supplementary Data 1). Following Harmony correction for section-to-section variation, this approach resulted in the identification of four clusters among L4 barcodes. A similar approach was implemented using P4 and P7 scRNA-seq data (Supplementary Fig. 2e–h). Here, gene signature scores for hollow- and septa-related genes facilitated clustering of L4 cells, with two distinct populations that displayed reciprocal marker scores. Using differential gene expression between putative hollow and septa clusters in the spatial and scRNA datasets, we amended our overall candidate list with additional screening for hollow and septa markers (Supplementary Data 2).

### TCA-dependent timing of whisker-related gene expression patterns

With these complementary approaches, we identified several genes that displayed reciprocal patterns of expression in L4, corresponding to the barrel hollow and septal compartments (Fig. 2a and Supplementary Fig. 1d–g). Moreover, these complementary, grid-like patterns were differentially regulated over the period of barrel formation in the first postnatal week. While we identified some genes that displayed a general L4 expression pattern (e.g., *RORβ*, *Dbp*), certain hollow markers followed a distinct time-course of expression that appeared spatiotemporally aligned with TCA innervation of barrel cortex, as previously described[29]. Low/undetected expression of these genes (e.g., *Btbd3*, *Astn2*, *Gpr158*, *Ier5*, *Dapk2*) was observed at birth, indicating that these are not solely general markers of L4 neurons. Instead, such genes became upregulated at P2 along with TCA innervation (as visualized with vGlut2 immunostaining), with whisker-related patterns emerging by P4 and becoming mostly formed within barrel hollows and excluded from septa regions by P7. As such, we regarded these genes as putative hollow-specific markers.

By contrast, septa-related genes displayed an overall broader pattern of expression in the cortical plate compared to hollow-related

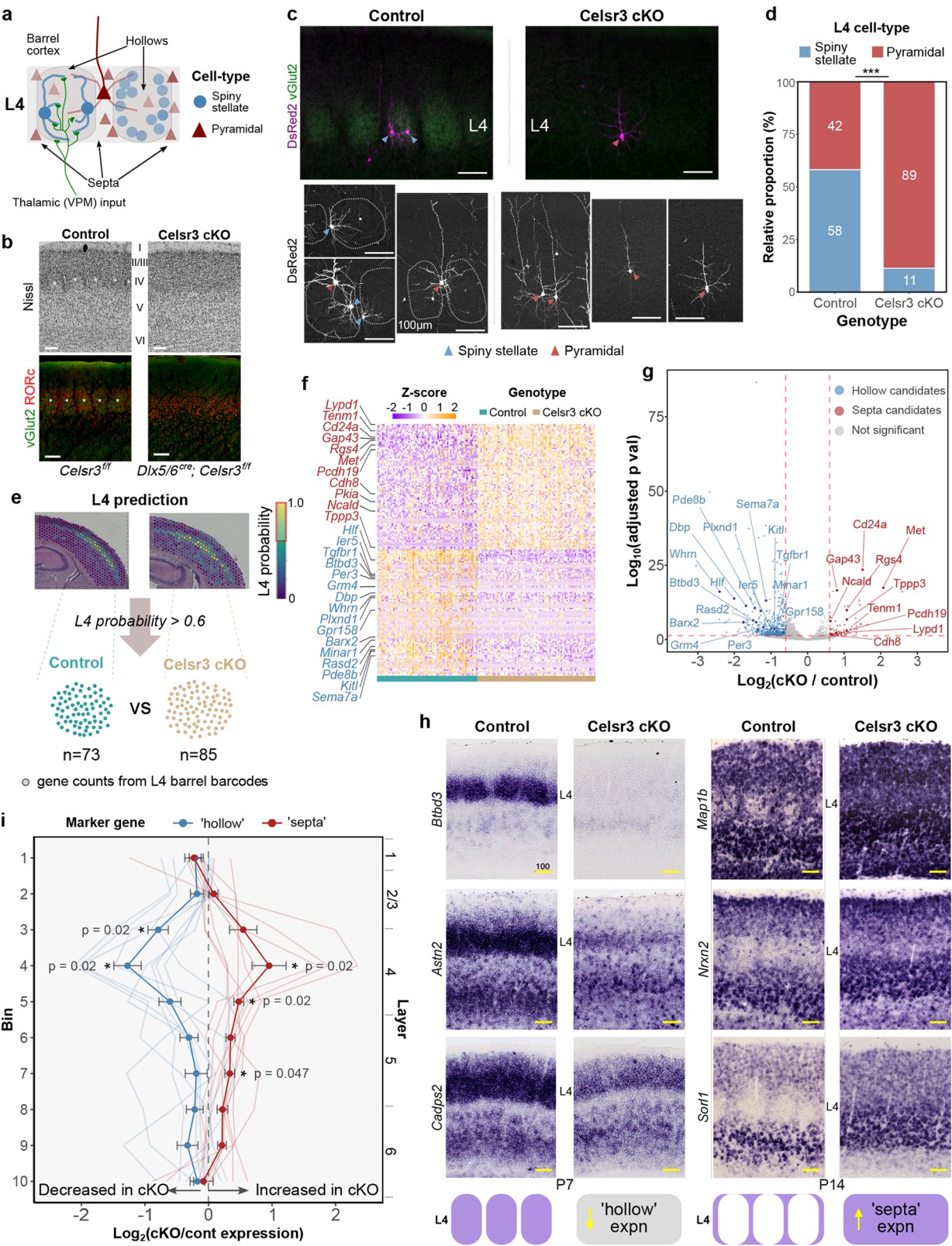

genes (Fig. 2b and Supplementary Fig. 1). These genes were widely expressed at birth across several brain regions and layers of the developing neocortex. While early expression overlapped with immunostaining for L4 marker, RORβ, it did not display the periodic restricted patterns found in mature barrels. Coincident with the increasing innervation and clustering of TCA input (P2–P4), we observed a specific reduction within L4 barrel cortex for these septa-

related genes, accompanied by a reduction in overlap with RORβ (Fig. 2b). After barrel formation is largely mature (P6), the broad expression of these markers was maintained in both infra- and supra-granular layers of the cortex, however, became restricted to the septa regions in L4 barrel cortex, observable as an inter-barrel pattern.

We further investigated how barrel gene expression may be regulated by activity from external inputs, utilizing two mouse models

**Fig. 1 | Spatial transcriptomics profiling of stellate-less somatosensory cortex in Celsr3 cKOs. a** Schematic illustrating the cytoarchitecture of mouse L4 barrel cortex, depicting spiny stellate neurons (blue circles) in hollow and pyramidal neurons (red triangles) in septal compartments. **b** In Nissl and RORc-immunostained sections, barrel clusters (asterisks) are absent in Celsr3 cKO somatosensory cortex with no thalamocortical innervation (vGlut2 staining). **c** Top, widefield fluorescent images of L4 neurons in "barrel" cortex visualized by sparse DsRed2 labeling. Bottom, representative confocal images showing spiny stellate (blue arrowheads) and pyramidal (red arrowheads) morphologies. In controls, barrels were outlined using DAPI and vGlut2 staining of TCA input (dotted lines). **d** Quantification of spiny stellate and pyramidal neurons in L4 somatosensory cortex. Asterisks indicate ***$p = 0.0002$, generalized linear mixed-effects model with parametric bootstrap test ($n = 162$ neurons, 19 control brains, $n = 185$ neurons, 8 Celsr3 cKO brains). **e** Spatial transcriptomes of Control and Celsr3 cKO somatosensory cortex were profiled at P7, with L4 "barrel" barcode positions predicted using annotated P7 scRNA-seq data. High-probability (>0.6, red outline) L4 spots

were selected, gene counts from each L4 barcode grouped by genotype, and differential gene expression was performed with DESeq2. **f** Heatmap displaying Z-scores for differentially regulated genes in L4 Celsr3 cKO. **g** Volcano plot displaying differential barrel gene expression between Celsr3 cKO versus control. Differentially regulated L4 barrel genes were determined using FDR < 0.05 and absolute $Log_2$(cKO/control) >0.6 as cut-offs (dashed red lines) using a two-sided Wald test with Benjamini–Hochberg correction. Screened genes are annotated in (**f**) and (**g**). **h** ISH of candidate hollow- (left) and septa-related (right) genes in Control and Celsr3 cKO somatosensory within P7/P14 coronal sections. **i** Quantification of mean hollow- and septa-related expression patterns. Relative expression (LogFoldChange: cKO vs control) was averaged across the cortex for hollow and septa genes (displayed are means ± SEM of 9 hollow- and 9 septa-related genes; asterisks indicate *$p < 0.05$, one-sided Wilcoxon Rank Sum test, $H_0$: μ<−0.1 or μ>0.1, multiple comparison correction using the Holm method). All scale bars, 100 μm. Source data are provided as a Source data file.

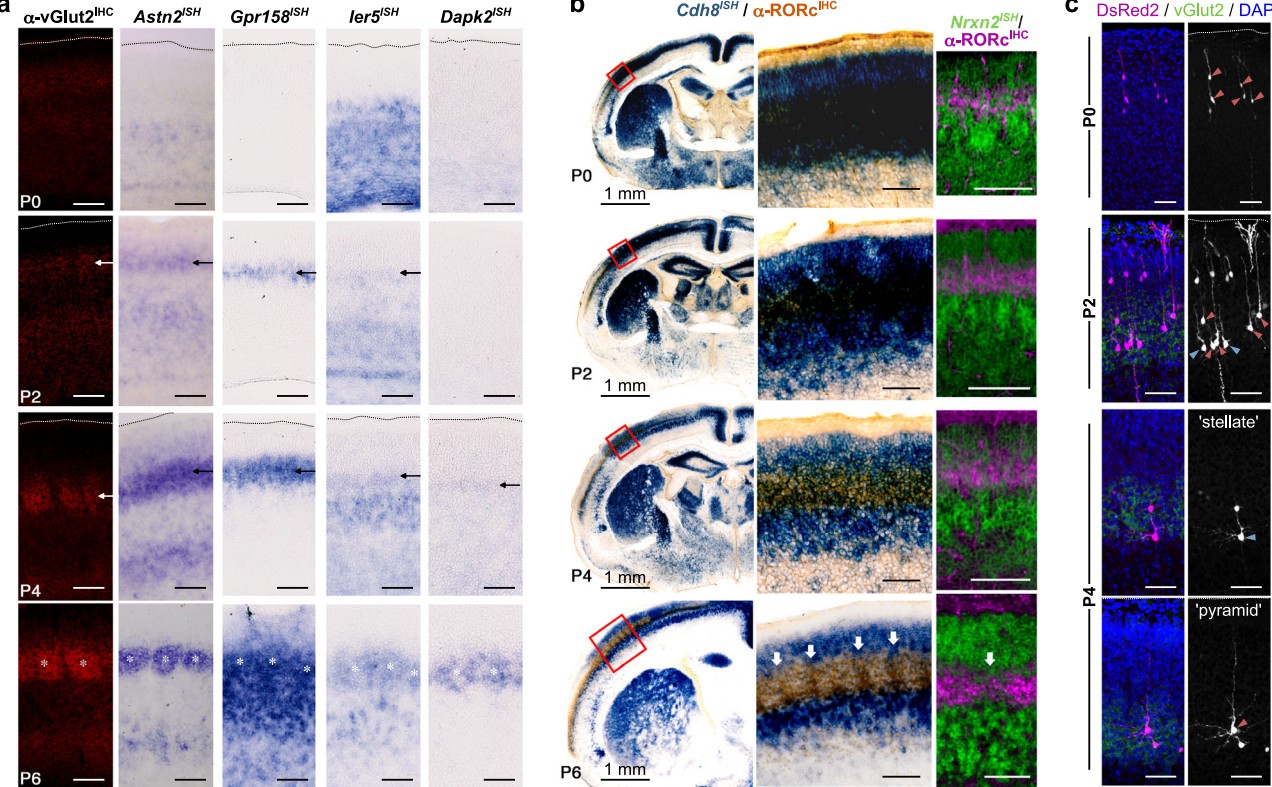

**Fig. 2 | Timing of reciprocal hollow- and septa-related gene expression in developing L4 barrel cortex is coincident with TCA innervation. a, b** A developmental time course of barrel formation was followed from P0 (top), up to the clear formation of barrel structures by P6 (bottom). **a** Specific hollow-related gene expression is up-regulated at P2 with onset of TCA innervation (indicated by arrows), as outlined by vGlut2 immunostaining (left column). This expression adopts a clustered barrel pattern by P6 (asterisks). **b** Septa-related gene expression (*Cdh8* and *Nrxn2* are shown) is broadly expressed in the developing neocortex at birth (top). Expression within the RORc⁺ L4 barrel region is progressively restricted upon TCA innervation, with inter-barrel patterns evident by P6 (arrows). Middle panels are higher magnification images of red outlined regions. **c** Appearance of spiny stellate neurons in barrel cortex is also associated with TCA innervation.

Sparse-labeling of L4 neurons with DsRed2 was performed by IUE at E13.5. At P0, L4-destined neurons are found with an immature morphology possessing an apical process that extends to the pial surface (blue arrowheads). At P2, while most neurons appear to have an immature morphology and an extended apical dendrite, some neurons appear to have shorter apical dendrites (blue arrowheads). By P4, a clear distinction between spiny stellate (blue arrowhead) and pyramidal morphologies (red arrowhead) could be made, with progressively more spiny stellate neurons found at the edge of barrels (visualized by vGlut2 staining), having a shorter apical dendrite and asymmetrical dendritic fields directed toward the center of TCA clusters. Unless indicated, scale bars are 100 μm. Dotted lines outline the pial surface.

with impaired TCA neurotransmission (Supplementary Fig. 3). In Sert^Cre; Rim1/2 double KO (dKO) brains[35], a targeted disruption of calcium-dependent synaptic release in thalamic neurons did not give rise to observable changes in hollow (*Btbd3*) or septa (*Cdh8*) gene expression, being found similar in appearance and levels to control brains. By contrast, with an inducible tetanus toxin (TeNT) model, a

more severe loss of TCA neurotransmission in Sert^Cre; TeNT animals[36,37] was associated with little to no *Btbd3* expression and upregulation of *Sorl1* specifically in L4. Interestingly, clustering of TCA inputs was also disrupted in these animals (Supplementary Fig. 3c–f), while is largely normal in Rim1/2dKOs[35]. While it is difficult to ascertain the exact role that activity has here in driving gene expression changes, our findings

here may suggest an upstream role in TCA targeting/clustering which may contribute to the differing phenotypes of the TCA activity mutants we investigated.

Notably, the morphological features of L4 barrel neurons developed along a similar time-scale as TCA innervation (Fig. 2c), as previously reported[18]. In addition, the respective downregulation and upregulation of hollow- and septa-related genes that we confirmed in Celsr3 cKOs not only reflected the change in L4 cell-types, but also resembled the profile of expression (low barrel/high septa gene expression) found in the immature cortex prior to TCA innervation (compare Figs. 1g and 2b). Taken together, these data highlight the instructive role of TCA input, required to upregulate expression of a set of genes that appear to be associated with spiny stellate neurons found within barrels. At the same time, a corresponding downregulation of default pyramidal/septa-related gene expression suggests that these concerted transcriptional events may account for the development and morphological/functional differences between barrel hollow and septal neurons.

### Smad7 is a septa-related gene that can repress gene expression and morphology of barrel compartments

In surveying the list of genes that display expression relating to either hollow or septa compartments, we hypothesized that certain genes may have a developmental role in barrel formation. Here, we focused on the TGF-β family whose members appeared to be expressed and suggested potential interactions within the thalamocortical pathway (Supplementary Fig. 4). Among the TGF-β family receptors, we confirmed several members as having barrel-related expression patterns (Tgfbr1, Tgfbr3, and Acvr1b) (Fig. 3a), while TGF-β signaling ligands (Gdf11, Inhbb, Tgfb1, and Tgfb3), were expressed in VPM thalamus (Fig. 3b). Given that Smad7, a potent inhibitor of TGF-β signaling[38], also displayed a mostly septa-like developmental pattern of expression in barrel cortex and was upregulated in Celsr3 cKO L4 S1 (Fig. 3c, d and Supplementary Fig. 2d), we tested whether its overexpression would affect gene expression and spiny stellate fate in the barrel cortex. Targeted overexpression of Smad7 (Smad7[OE]) by IUE of neural progenitors destined for L4 barrel cortex led to a reduction in barrel hollow gene expression and conversely, an upregulation of septa gene expression (Fig. 3e, f and Supplementary Fig. 5a). We found no major changes to the position of cortical layer marker expression (Tbr1, Fezf2, Etv1, Satb2) with Smad7[OE] (Supplementary Fig. 5b).

In contrast, the above changes were not observed with overexpression of nuclear-localized Smad7 (Smad7-NLS) or a Tgfbr1 binding mutant[39], Smad7[K312E,K316E], suggesting the ability of Smad7 to regulate barrel gene expression is dependent on its binding and regulation of TGF-β receptors[40], and not directly at the level of transcription[41] (Supplementary Fig. 5c, d). To demonstrate a possible involvement of Tgfbr1 in the barrel pathway, we performed knockdown experiments with IUE of Tgfbr1-shRNA constructs targeted to L4 neurons (Tgfbr1[KD]). Here, while we found a significant reduction in the size of barrels and changes to barrel gene expression with Tgfbr1[KD], this phenotype was not as severe compared to Smad7[OE], indicating that other family members or interacting factors may contribute to the Smad7[OE] phenotype (Supplementary Fig. 5e–g). Together, these data suggest a form of TGF-β signaling, involving the Tgfbr1 receptor, may have important roles in specific thalamocortical interactions. Further investigation, however, will be required to understand the exact molecular pathway involved in regulation of L4 cell fate by Smad7.

Consistent with changes in gene expression, we also found a shift in the population of L4 neurons in barrel cortex with Smad7[OE]. Sparse-labeling was performed by IUE of Cre-dependent HA-tagged Smad7 and/or EYFP plasmids (Fig. 3g, h). Whereas the population of control[EYFP] neurons consisted of both spiny stellate and pyramidal morphologies, the majority of Smad7[OE] neurons possessed a prominent apical dendrite/pyramidal-like morphology (94% vs 42% in

controls; $\beta_{Smad7OE} = 3.06$; s.e. $= 0.58$; $\sigma_{animalID} = 0.26$; generalized linear mixed-effects model with parametric bootstrap test, $p = 0.0002$) with few spiny stellate neurons observed (6% vs 58% in controls).

Using a transgenic mouse strain that allowed Cre-dependent Smad7 overexpression (conditional Smad7[Tg]), we observed a similar effect of Smad7[OE] on L4 gene expression and morphology within NEX[Cre]; Smad7[Tg] barrel cortex (Supplementary Fig. 6). While the barrels still form in this mouse, reciprocal changes in hollow and septa gene expression were observed. The overall size of S1 as well as the posteromedial barrel field (PMBF) was significantly smaller in Smad7[Tg] animals, with the anterior barrels also less defined (Supplementary Fig. 6a–c). Furthermore, the percentage of pyramidal neurons was significantly increased in Smad7[Tg] barrel cortex (Supplementary Fig. 6d). Altogether these effects appeared to be consistent and dose-dependent, however, were not as dramatic as those changes seen with higher levels of expression achieved with IUE.

With further investigations of Smad7[OE], no observable differences in the generation and radial migration of transfected neurons were found, with the number and position of Smad7[OE] comparable to controls at P0 (Supplementary Fig. 7b). L4 RORβ expression was still present at its normal radial position in the cortex, however, its distinct clustering into barrel-like regions was no longer observed. This lack of barrel arrangement was further evident with nuclear DAPI staining, while vGlut2 clusters and CO staining showed a similar alteration to overall barrel structure (Supplementary Fig. 7a, b). We further noted a tendency for Smad7[OE] neurons to be located around and in between smaller vGlut2+ TCA clusters in what is likely to be the septa region (Supplementary Fig. 7b). While smaller TCA clustering in Smad7[OE] brains may contribute to this, it is possible that while these neurons do not respond to TCA-derived cues as would spiny stellate neurons, there may be additional signals that cause local repulsion of septal neurons by TCA clustering. Alternatively, these neurons may become displaced by remaining spiny stellate neurons that do not express Smad7.

Quantification of cell numbers in electroporated regions (Supplementary Fig. 7c) indicated that Smad7[OE] led to moderate reductions in L4 density (87% of controls) and L4 RORb+ density (67% of controls). Thus, it is important to note that Smad7[OE] may lead to a selective loss of neurons destined to become spiny stellate neurons. However, the magnitude of reductions we observe here are not comparable to the majority spiny stellate population seen in controls (being 60–70% of barrel neurons), nor the magnitude of changes to gene expression and morphology, suggesting that additional factors (e.g., TCA clustering) may be contributing to these differences.

In focusing our analyses on those neurons overlapping with regions of vGlut2 immunostaining, our data suggest that despite the presence of TCAs within L4 barrel cortex, Smad7[OE] neurons can no longer respond appropriately to thalamic cues that drive formation of barrel circuits, thus unable to acquire gene expression, morphology, and cellular reorganization associated with spiny stellate neurons.

### Increased contralateral projections from septal-type L4 neurons with Smad7[OE]

Since Smad7[OE] neurons in barrel cortex had acquired the morphology and gene expression of L4 pyramidal neurons, we investigated whether they also adopted functional connections associated with these septal-type neurons. The main callosal projection from S1 arises from L2/3 and L5 and follows a distinct developmental trajectory that is dependent on appropriate sensory input[42–45]. Interestingly, in S1 there appears to be a distinct developmental program for the elimination of the widespread, transient callosal projections from L4[46]. Given the influence of TCA input in this L4-specific process and previous reports describing a minor septa-related L4 projection in adult rats[47,48], we investigated whether this population would be impacted by Smad7[OE].

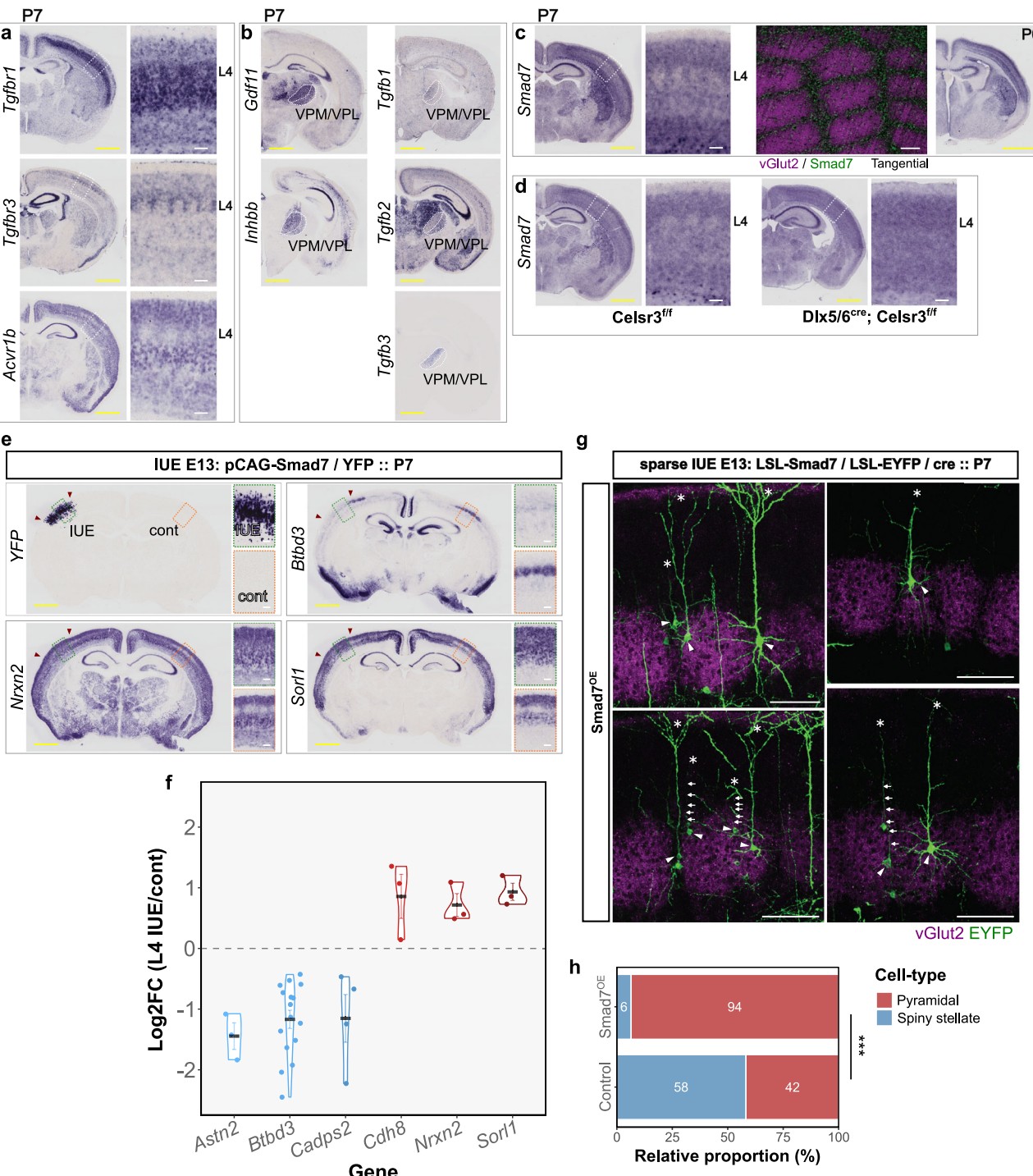

**Fig. 3 | Overexpression of septa-related gene, *Smad7*, recapitulates gene expression and morphology changes seen in Celsr3 cKO barrel cortex.** TGF-β family members display specific expression of receptors (**a**) and ligands (**b**) within the barrel thalamocortical pathway. **c** Inhibitor of TGF-β signaling, *Smad7*, is expressed in barrel cortex in a septa-related pattern. **d** *Smad7* is upregulated in L4 of Celsr3cKO S1 cortex. **e** Smad7/EYFP overexpression (Smad7$^{OE}$) leads to down-regulation of hollow-related genes (*Btbd3*), and upregulation of septa-related genes (*Nrxn2* and *Sorl1*). Images are from a representative electroporated brain (P7) with RNA ISH for the indicated genes performed on serial sections. The extent of transfection is indicated by red arrowheads. Gene expression within electroporated areas (IUE) are outlined in green, while the un-electroporated barrel cortex (cont) is outlined in orange. **f** Quantification of expression changes in Smad7$^{OE}$ brains.

Log$_2$FoldChange was calculated for L4 expression intensity of indicated genes in the IUE vs Cont hemisphere. Displayed are group means ± SEM. Astn2: *n* = 3; Btbd3: *n* = 16; Cadps2: *n* = 4; Cdh8: *n* = 3; Nrxn2: *n* = 3; Sorl1: *n* = 3 brains. **g** Morphology of L4 Smad7$^{OE}$ neurons which were sparsely transfected with Cre-dependent Smad7/EYFP (LSL=LoxP-STOP-LoxP). A prominent apical dendrite (asterisk) was found on most Smad7$^{OE}$ neurons (arrowheads). In some images, arrows indicate the path of the apical dendrite. **h** Quantification of spiny stellate versus pyramidal neuron morphology in sparsely labeled Smad7$^{OE}$ neurons in L4, compared to control overexpression. Asterisks indicate ***$p$ = 0.0002, generalized linear mixed-effects model with parametric bootstrap test (*n* = 158 neurons from 19 control brains, *n* = 62 neurons from 5 Smad7$^{OE}$ brains). Scale bars, 1 mm (yellow) and 100 µm (white). Source data are provided as a Source data file.

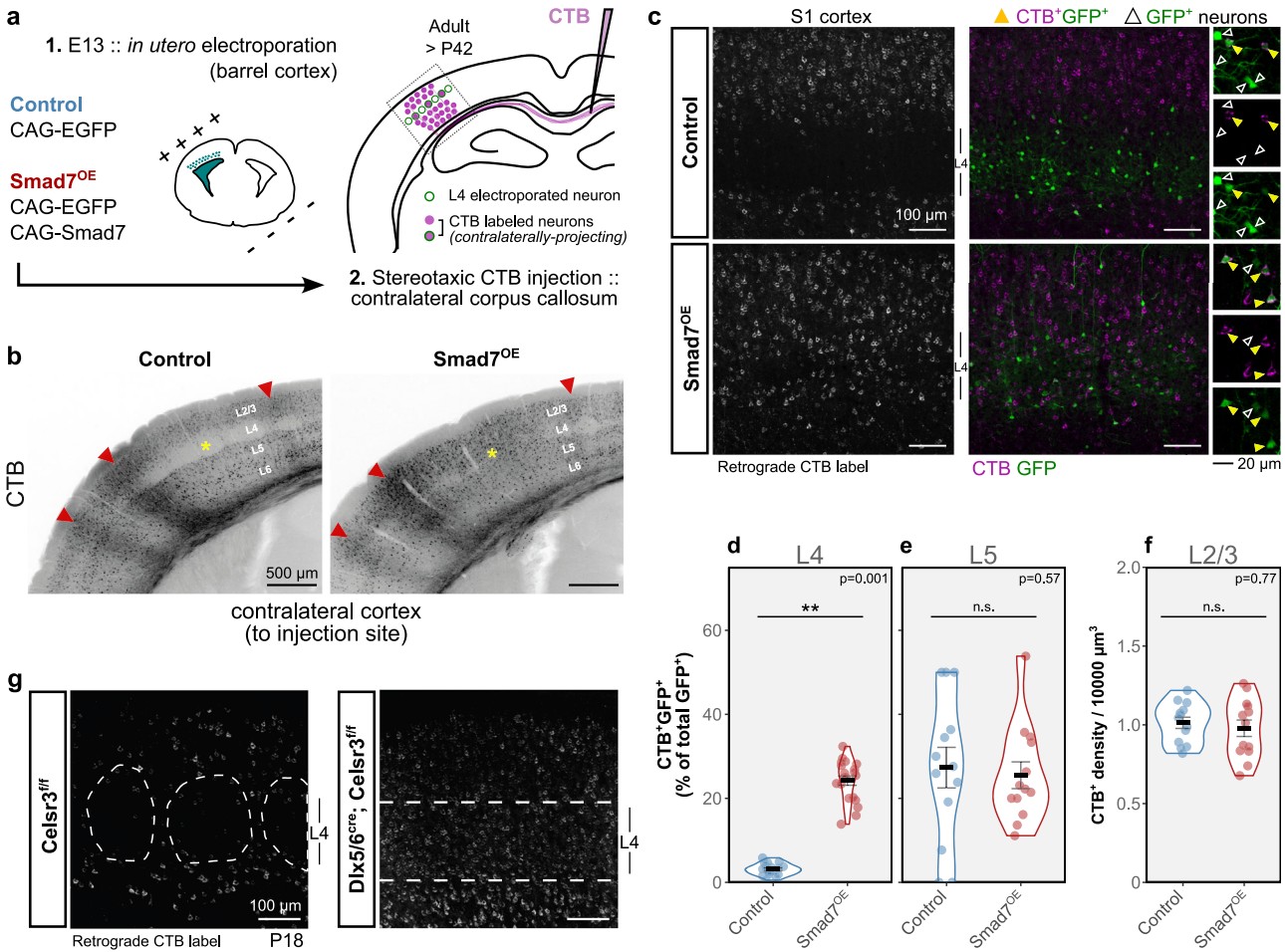

**Fig. 4 | Output connectivity of Smad7OE neurons. a** Schematic illustrating stereotaxic CTB injections into the corpus callosum to label contralaterally-projecting neurons in barrel cortex of adult control and Smad7OE brains. **b** Retrogradely labeled (CTB+) neurons in the cortex contralateral to the injected corpus callosum. A high degree of retrograde label is found in supra- and infra-granular layers in both genotypes. In adults, L4 is sparsely labeled in controls compared to Smad7OE (yellow asterisk). Anterograde labeling can also be observed as diffuse axonal signals (red arrowheads). **c** Retrogradely-labeled (CTB+) transfected L4 (GFP+) neurons in barrel cortex, contralateral to the injected area. Within L4, sparsely-distributed CTB+ neurons were labeled in controls. With Smad7OE, an increased amount of CTB labeling was observed throughout L4. Example regions showing double-positive CTB+/GFP+ (yellow triangles) and CTB-GFP+ (open triangles) transfected neurons are shown on the right. **d**–**f** Quantification of CTB+ neurons displayed as percentage of double-positive CTB+/GFP+ (of total number of electroporated GFP+ neurons) in L4 (**d**), L5 (**e**), and CTB+ density in L2/3 (**f**). Violin plots show distributions from 6 sections (on average) from 3 brains per genotype. Group means are displayed with error bars representing SEM. Asterisks indicate **$p = 0.001$, linear mixed-effects model with parametric bootstrap test ($n = 19$ sections from 3 control brains, $n = 20$ sections from 3 Smad7OE brains). **g** Labeling of retrogradely projecting neurons in somatosensory cortex of Celsr3 control (flox/flox) and cKO brains at P18. Source data are provided as a Source data file.

Accordingly, using targeted injections of retrograde tracer, cholera toxin subunit b (CTB), into the contralateral corpus callosum (Fig. 4a), we also observed that while CTB-labeled neurons in adult control (E13 IUE: GFP) mice were abundant in supra- and infragranular layers of barrel cortex, relatively few retrogradely labeled neurons were identified in L4, with those being situated predominately in septa regions around vGlut2+ TCA clusters (Fig. 4b, c). By contrast, we found that Smad7OE (E13 IUE: Smad7/GFP) resulted in a greater number of L4 contralaterally projecting neurons in barrel cortex, consistent with there being an increased pyramidal/septal population in these mice (Fig. 4b, c). Normalizing for the number of electroporated cells in L4, a significantly higher proportion of CTB+GFP+ neurons (as a percentage of L4 GFP+ neurons) was observed for Smad7OE, compared to GFPOE (L4 CTB+GFP+ % ± SEM: GFPOE, $3.10 ± 0.31$; Smad7OE, $24.20 ± 1.09$; $\beta_{Smad7OE} = 21.10$; s.e.$=1.16$; $\sigma_{animalID} = 3.78$; linear mixed-effects model with parametric bootstrap test, $p = 0.001$). Despite our transfections mostly targeting L4, GFP+ cells were also found in L5 for both groups, resulting from the larger windows of transfection for IUE. Although more variable, we quantified similar proportions of L5 CTB+GFP+

neurons between groups (L5 CTB+GFP+ % ± SEM: GFPOE, $27.30 ± 4.83$; Smad7OE, $25.50 ± 3.18$), suggesting that callosal projections were specifically enhanced in L4 with Smad7OE. Furthermore, the degree of contralateral projections observed in L5 appeared similar to L4 of Smad7OE animals. No differences in the density of labeled L2/3 neurons were found (Fig. 4d–f).

We found a similar increase in L4 contralateral projections with corresponding labeling experiments in Celsr3 cKO animals, which we performed at P16-P18 due to their poor survival past three postnatal weeks. Here, a striking difference was observed between control animals and stellate-less Celsr3 cKOs, in which, corresponding to the default pyramidal identity we found in the absence of TCA innervation, a large degree of retrograde CTB labeling was present throughout the S1 region (Fig. 4g). Thus, maintaining pyramidal identity in L4 neurons by Smad7OE or in the Celsr3 cKO resulted in increased contralateral projections associated with septal circuits.

Notably, the establishment of local connectivity by TCA innervation is required for the elimination of most callosal projections from L4, which is consistent with our findings in the Celsr3 cKO. It remains to

be determined whether the changes we see in our models are due to an overall deficit in activity affecting callosal development[44,46,49]. Given that activity-dependent refinement and the elimination of L4 callosal projections seem to operate at later timepoints to the establishment of barrel cell identity[46], our data highlight the importance of early acquisition of spiny stellate fate in these processes. While the precise targets and functional relevance of this L4 contralateral projection remains to be determined, these data underscore a role for TCA innervation in shaping the degree of L4 callosal outputs, through its impact on driving spiny stellate fate and barrel circuits from its initial pyramidal cortical state in L4.

## Altered input connectivity of Smad7[OE] neurons is similar to pyramidal neurons in L4 barrel cortex

The morphological differences between L4 spiny stellate and pyramidal neurons are likely to determine which inputs they receive. The widespread dendritic field and no apical dendrite of septal (pyramidal) neurons may distinguish them from those in the barrel (spiny stellate) by facilitating more diverse and long-range connections. In the barrel cortex, whereas L4 spiny stellate neurons receive very little excitatory cortical input from outside its home barrel, intracortical connections to L4 pyramidal neurons are indeed more widespread[4,10,50]. We therefore asked whether local inputs to L4 neurons in barrel cortex were also altered by Smad7[OE], utilizing photostimulation-mediated glutamate uncaging to map intracortical connections in cortical slice (P13-P19) preparations (Fig. 5a–f). In control (naïve wildtype) slices, recorded L4 neurons were characterized as spiny stellate neurons, with cortical inputs coming mostly from its home barrel and within L4. In contrast, Smad7[OE (Smad7/CyRFP)] neurons received significantly more inputs from infragranular layers, both within (intracolumnar) the recorded column (L5a and L5b) and from neighboring (transcolumnar) barrel columns (L5a). This connectivity pattern of Smad7[OE] neurons appeared similar to that reported for L4 pyramidal neurons in barrel cortex[10].

Our findings suggested that Smad7[OE] neurons are likely to have a more integrative role in barrel cortex, adopting the multi-whisker receptive fields of septal pyramidal neurons. We further investigated this beyond intracortical connectivity with monosynaptic rabies virus tracing to determine whether there were any systematic differences in input connectivity to L4 Smad7[OE] neurons (Fig. 5g–k). To prime neurons for rabies virus infection, we sparsely transfected Cre-recombinase along with Cre-dependent plasmids (Smad7 and/or BFP–histoneGFP–TVA[66T]–Glycoprotein[N2c]) into L4 barrel cortex using IUE[51]. After stereotaxic injection of pseudotyped rabies virus (EnvA-CVS-N2c[ΔG]-mCherry) into adult L4 barrel cortex, we performed brain-wide tracing of presynaptic inputs (mCherry[+]) to Control and Smad7[OE] starter neurons (mCherry[+]GFP[+]), whose locations were registered to the Allen Mouse Brain Reference Atlas (CCFv3). Following normalization to the number of L4 starter neurons (calculated as index of connectivity: IOC = number of presynaptic neurons in brain area/ starter neurons) we found further evidence of increased intracortical connectivity within the barrel field (Fig. 5i; $p < 0.05$, Wilcoxon Rank Sum test), as suggested by our previous findings. Here, IOC of L2/3, L4, and L5 in S1 barrel field was significantly higher for Smad7[OE] starter neurons, compared to control neurons ($p < 0.05$, Wilcoxon Rank Sum test). Moreover, rabies virus tracing revealed a higher degree of inputs to Smad7[OE] starter neurons from POm thalamus (Fig. 5j, k; $p < 0.05$, Wilcoxon Rank Sum test), consistent with the thalamic connectivity associated with their maintained septal/pyramidal identity. This was further linked with a decrease in the proportion of VPM-versus-POm inputs onto Smad7 starter neurons ($p = 0.06$), resulting in a ratio approaching 1. This may indicate that mixed whisker-derived inputs are received by Smad7[OE] neurons. We found no further significant differences in Smad7[OE] neuron presynaptic connectivity with other cortical or subcortical areas (Supplementary Fig. 8).

## Whisker-evoked activity in barrel cortex is altered by Smad7[OE]

To evaluate how sensory information is processed within the barrel cortex of Smad7[OE] mice, in which the population of L4 neurons (hollow vs septa) that receive information from TCA inputs is altered, we first looked at c-fos expression as an initial readout of neuronal activity downstream of whisker activation. After IUE at E13, adult control[EYFP] and Smad7[OE (Smad7/EYFP)] mice were whisker-trimmed, leaving a single whisker (C2) intact, contralateral to the electroporated cortex (Fig. 6a). After exploration of a novel texture environment, this remaining whisker led to induction of c-fos expression that was mostly confined to its corresponding cortical barrel in control animals. This contrasted with neighboring barrels that were deprived of direct whisker-related information and display a much lower level of c-fos, with sparse expression in septa regions resulting in a subtle net-like appearance in overlayed images (Fig. 6b). By contrast, we found c-fos expression in Smad7[OE] mice was no longer restricted to the intact C2-whisker barrel, rather appeared more variable and widespread in neighboring barrel regions (Fig. 6b–d).

Using widefield calcium imaging of barrel cortex, we found similar differences in vivo for whisker-evoked activity (Fig. 6e–m). Cortical responses to single whisker air-puff stimulation were imaged in S1 of lightly anesthetized adult Thy1-GCaMP6f mice, following IUE of Smad7 (GCaMP6f[Smad7-OE]) and/or CyRFP control (GCaMP6f[Control]) plasmids to target L4 barrel cortex. The peak response to whisker stimulation was not significantly different between GCaMP6f[Smad7-OE] and GCaMP6f[Control] mice (Fig. 6h). We found, however, significant differences in the spatiotemporal extent of whisker-evoked activity in GCaMP6f[Smad7-OE] barrel cortex, occurring over a significantly larger area and of longer duration, compared to controls (Fig. 6f–m). Together, these data suggest a decreased prevalence of spiny stellate neurons results in more widespread activation of barrel cortex via increased inputs onto septal-type neurons.

## Smad7[OE] mice display an impaired ability to discriminate between textures

The segregation of individual whisker input by the lemniscal/spiny stellate circuit is likely to form the basis of fine tactile discrimination afforded by the whisker somatosensory system. We therefore employed a novel texture discrimination (NTD) task[52], to test the ability of Smad7[OE] mice to discern between two different textures. This whisker-dependent behavioral task utilizes the tendency for mice to explore novelty in their environment, introduced here in the form of two distinct sandpaper textures (Fig. 7a, b). Here, we paired unilateral Smad7[OE] with unilateral whisker-trimming (ipsilateral to transfection) to increase the whisker-dependency of this task. This also allowed us to specifically test the Smad7-transfected cortex, while trimming of whiskers contralateral to the transfected cortex provided an additional control for Smad7[OE] mice (Smad7[OE-control]). After habituation to an empty test arena, adult mice were presented with two texture panels of the same coarseness (A[rough] vs A[rough]) during a pre-test session. Following a brief (5 min) home cage recovery period, NTD was subsequently tested by introducing a distinct texture into the arena (A[rough] vs B[fine]). Mice from both control groups (EYFP[OE] and Smad7[OE (whisker-trim control)]) accordingly spent more time exploring the novel texture B, compared to the initial, familiar texture A (Fig. 7c). As a measure of discriminatory ability, an NTD index was calculated for preference towards the novel texture (Fig. 7d). Whereas NTD indices for control mice were positive towards the novel texture (mean NTD index ± SEM: EYFP[OE], 0.31 ± 0.068; Smad7[OE-control], 0.20 ± 0.056), those for Smad7[OE] mice were significantly lower (ANOVA: $F_{2,29} = 13.92$, $p < 0.001$) and centered around zero (mean NTD index ± SEM: −0.13 ± 0.066; $p < 0.001$ and $p < 0.01$ Tukey HSD post-hoc test, Smad7[OE] compared to EYFP[OE] and Smad7[OE-control] groups, respectively). Given Smad7[OE] was targeted to L4 barrel cortex, impairments to memory formation, a requisite for

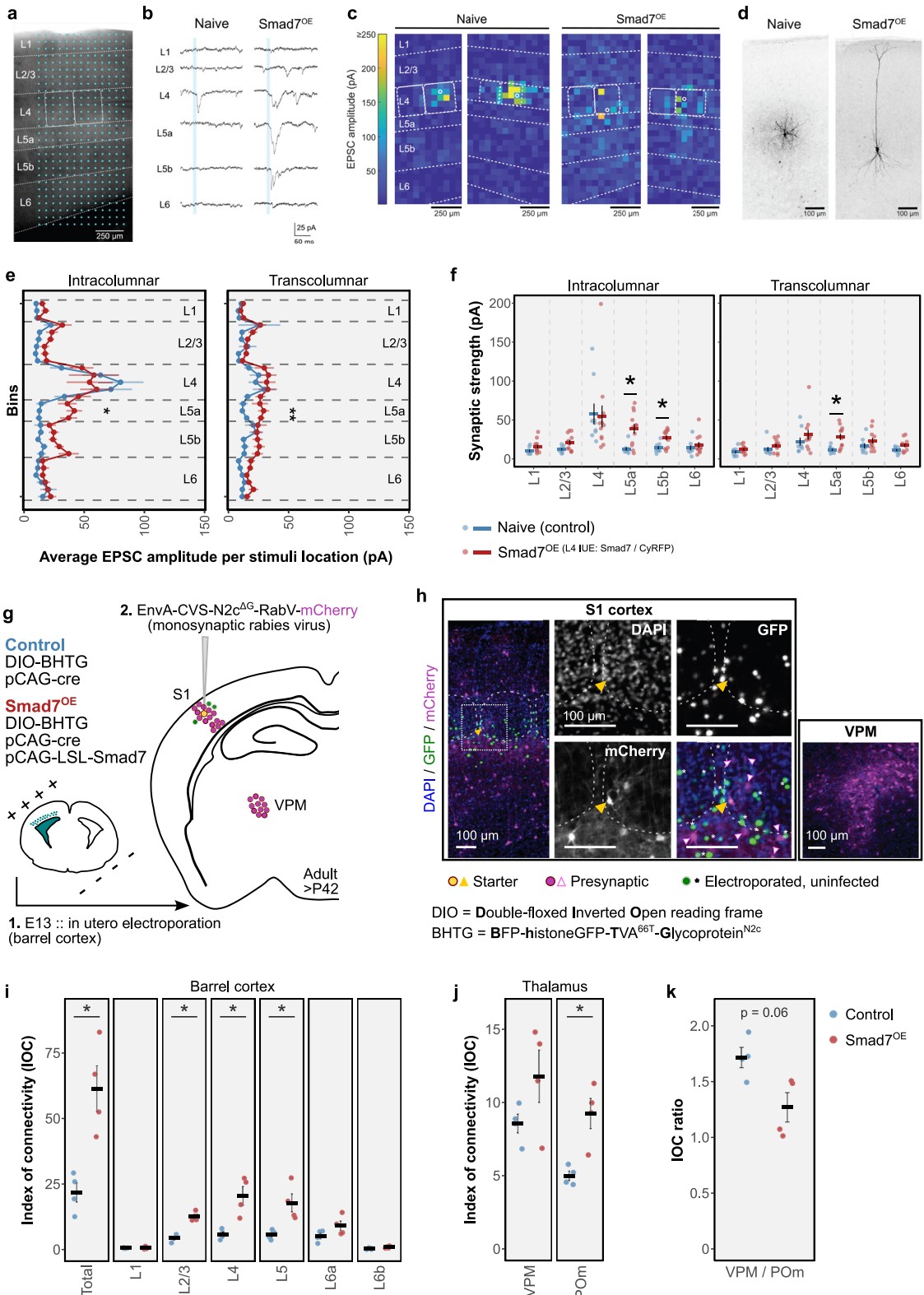

recognition of novel vs original textures, are less likely to explain these observations. Importantly, we found no gross differences in anxiety or exploratory behaviors, with total texture interacting times not significantly differing between control and Smad7[OE] groups (Fig. 7e; mean ± SEM: EYFP[OE], 52.0 ± 6.0 s, Smad7[OE], 44.2 ± 7.0 s; Smad7[OE-control], 59.5 ± 11.6 s; ANOVA: $F_{2,29} = 0.80$, $p = 0.46$). Taken together, our data suggest that the differences in NTD indices are likely due to an impaired ability to distinguish between distinct textures which result from the previously determined alterations to barrel cortical circuits.

## Discussion

An area of particular interest regarding cortical development is how converging inputs within the brain interact to shape the circuitry within their target area, a process which is likely to involve intrinsic genetic mechanisms and shaped by neuronal activity/experience. This

**Fig. 5 | Mapping input connectivity to L4 Smad7$^{OE}$ neurons using photo-stimulation glutamate-uncaging and monosynaptic rabies virus tracing.**
**a** Differential interference contrast image of barrel cortex slice (P13-P19) showing cortical layers and patch pipette positioned in L4. Dashed lines and rectangles indicate laminar borders and L4 barrels, respectively. Blue dots indicate photo-stimulation sites. **b** EPSCs evoked in L4-barrel neurons by photostimulation at representative sites in each cortical layer (L1-L6). Blue lines indicate the period of photostimulation. EPSCs were recorded from naive (left) and Smad7$^{OE}$ neurons (right). **c** Representative heatmaps displaying the sum of EPSC amplitudes evoked in L4 neurons by photostimulation at each site. Circles indicate location of recorded neurons. **d** Examples of biocytin-stained L4 neurons. **e** Averaged EPSC amplitude per stimulation site across binned locations within the home column of recorded cells (intracolumnar, left) and outside the home column (transcolumnar, right) in naive (blue) and Smad7$^{OE}$ neurons (red). **f** Strength of synaptic inputs in each stimulus layer. In (**e**, **f**), displayed are group means ± SEM. *$p < 0.05$, two-sided Welch's $t$ test with multiple comparison correction using the Holm method ($n = 9$

naive neurons, $n = 14$ Smad7$^{OE}$ neurons). **g** Schematic illustrating rabies virus-mediated labeling of presynaptic inputs to Smad7$^{OE}$ neurons. L4 neurons were sparsely transfected by IUE (green circles) with Cre-dependent plasmids encoding rabies starter construct (BHTG) ± Smad7. Adult animals were injected with pseudotyped rabies virus to induce trans-synaptic labeling from starter neurons (yellow circle), with presynaptic neurons (magenta circles) labeled with mCherry.
**h** Example of a starter neuron (yellow triangle) in L4 barrel cortex (left). Dashed box indicates area of magnified images shown on the right. Barrel outlines are indicated by dashed lines. Presynaptic neurons are indicated by magenta triangles; uninfected, electroporated neurons (green) are indicated by asterisks. Right panel, presynaptic neurons labeled in VPM thalamus. **i**, **j** Index of connectivity (IOC) for traced neurons in ipsilateral barrel cortex (**i**) and thalamus (**j**). **k** IOC ratio of VPM/POm thalamus. In (**i**–**k**), graphs are plotted with group means ± SEM. *$p = 0.03$, two-sided Wilcoxon Rank Sum test ($n = 4$ Control and $n = 4$ Smad7$^{OE}$ brains). Source data are provided as a Source data file.

may be especially pertinent in developing primary sensory cortices, where early activity is driven predominantly by bottom-up inputs conveying external information via projections relayed from the thalamus.

Here, we found a specific role for these extra-cortical inputs in driving cell-type diversity within L4 barrel cortex. The specification of spiny stellate neurons seems to be associated with a defined genetic pathway that emerges from an immature cortical state that is by default "pyramidal" in its expression profile and morphology. As such, the development and clustering of stellate neurons appears intimately tied to their inputs, whereas the septal compartments are comprised of L4 pyramidal neurons that do not receive whisker-specific inputs from VPM thalamus.

A key resource for this study was a mouse model harboring a conditional KO of Celsr3[31,53]. While this member of the protocadherin family has important functions in brain development, Celsr3 expression in the ventral telencephalon can regulate thalamocortical projections independent of its roles in cortical or thalamic neurons[31,54,55]. It is with this cell non-autonomous role which we aimed to test the scenario in which TCA innervation of the cortex is absent using a Dlx5/6$^{IRES-Cre}$ line, thus minimize intrinsic projection/target effects. It should be noted that due to abnormal axon pathfinding at the internal capsule, the major projections in the internal capsule (thalamocortical, corticothalamic, subcerebral) are disrupted in this animal[31]. While several studies have demonstrated specific roles for TCA input, this raises the indirect possibility that failure to establish corticofugal projections may disrupt barrel formation. This should be considered here, as well as whether changes in additional connections targeting barrel cortex also contribute to the phenotype seen in this mutant. It is a distinct possibility that development of the globus pallidus is also affected in this Celsr3cKO mutant, given the majority of its cells are derived from the MGE. How a possible GP deficit would directly impact the early development of barrel cortex is unclear given their indirect connectivity. It is likely, however, that it would further contribute to the pathfinding deficit of thalamocortical axons around the subpallium given its vicinity to the internal capsule/corridor cells[55–57]. Additionally, while interneuron migration and maturation in barrel cortex does not appear to be severely affected in this mouse[31], there is a likelihood, that without proper Celsr3 expression their function or integration into cortical circuits may be altered and have an impact on barrel formation. This should be an important consideration for the interpretation of our data since early interneuron populations appear to regulate aspects of cortical maturation[58,59].

We have identified several hollow-related genes that may act downstream of specific TCA input to drive L4 stellate fate. While certain markers (*Btbd3*, *Cadps2*, *Sema7a*,) were dependent on TCA innervation, manipulation of their expression is reported to influence dendritic orientation rather than spiny stellate fate, indicating

functional roles downstream of specification[27,29] (unpublished observations). Interestingly, this also appears to be the case for several reported genetic manipulations that affect barrel formation[35,60,61]. Thus, whether any barrel markers identified in this study are located upstream in this pathway remains to be determined, however, it is likely that factors regulating a transcriptional program driving spiny stellate-specific markers and repressing early pyramidal/septa-related expression are involved. While L4 marker, RORβ, has been reported to be necessary and sufficient for forming barrel cytoarchitectures[62,63], its L4 expression is present before TCA innervation and is also not specific to barrel cortex. It is likely, however, that specific thalamic input intersects with this genetic pathway to drive spiny stellate specification, consistent with the suggested role that this transcription factor has in promoting both thalamic innervation and subsequent refinement of cortical circuits[64].

While utilizing a spatial transcriptomics approach helped us to systematically identify L4 gene expression relating to barrel compartments, a lack of cellular resolution precluded a differentiation between cell-types, nor did we assess potential differences between pyramidal or star pyramidal neurons. Although our ISH screen helped to filter non-neuronal patterns of expression, whether other cell-types (e.g., inhibitory interneurons, astrocytes, microglia) are also contributing to patterns of barrel gene expression and its formation will be important to understand.

The identities of precise cues provided by TCA input to initiate cortical diversification are of particular interest. The specificity of certain spiny stellate-related genes (for example, Btbd3) in mouse barrel cortex, compared to visual cortex[29], may suggest the involvement of factors derived specifically from VPM thalamic axons (Supplementary Fig. 4). Indeed, TCA-derived molecular cues can regulate aspects of L4 formation[20], therefore how these elements might also contribute to spiny stellate specification will be of further interest. It is also clear that barrel formation and spiny stellate morphology can be influenced by activity[35,60,61,65,66]. In light of findings of the importance of prenatal thalamic spontaneous activity for barrel formation, manipulations to activity starting from these early timepoints may preclude understanding the direct role of neurotransmission at the time of spiny stellate acquisition. Thus, the exact role that activity has in determining spiny stellate fate and barrel gene expression, is currently hard to dissect. Our findings in the Sert$^{Cre}$; TeNT$^{Tg}$ mouse suggest that TCA neurotransmission is indeed important for regulating barrel gene expression and formation (Supplementary Fig. 3). Our observations in the Sert$^{Cre}$; Rim1/2$^{dKO}$ mouse, however, suggest that this activity relationship is perhaps more nuanced. In these mutants, a disruption in calcium-dependent synaptic release (70–90% reduction[67]) that results in "normal" barrel gene expression and structure is puzzling, but may either indicate a low threshold for neurotransmission in controlling these events or reflect appropriate cues being provided by largely normal innervation seen in

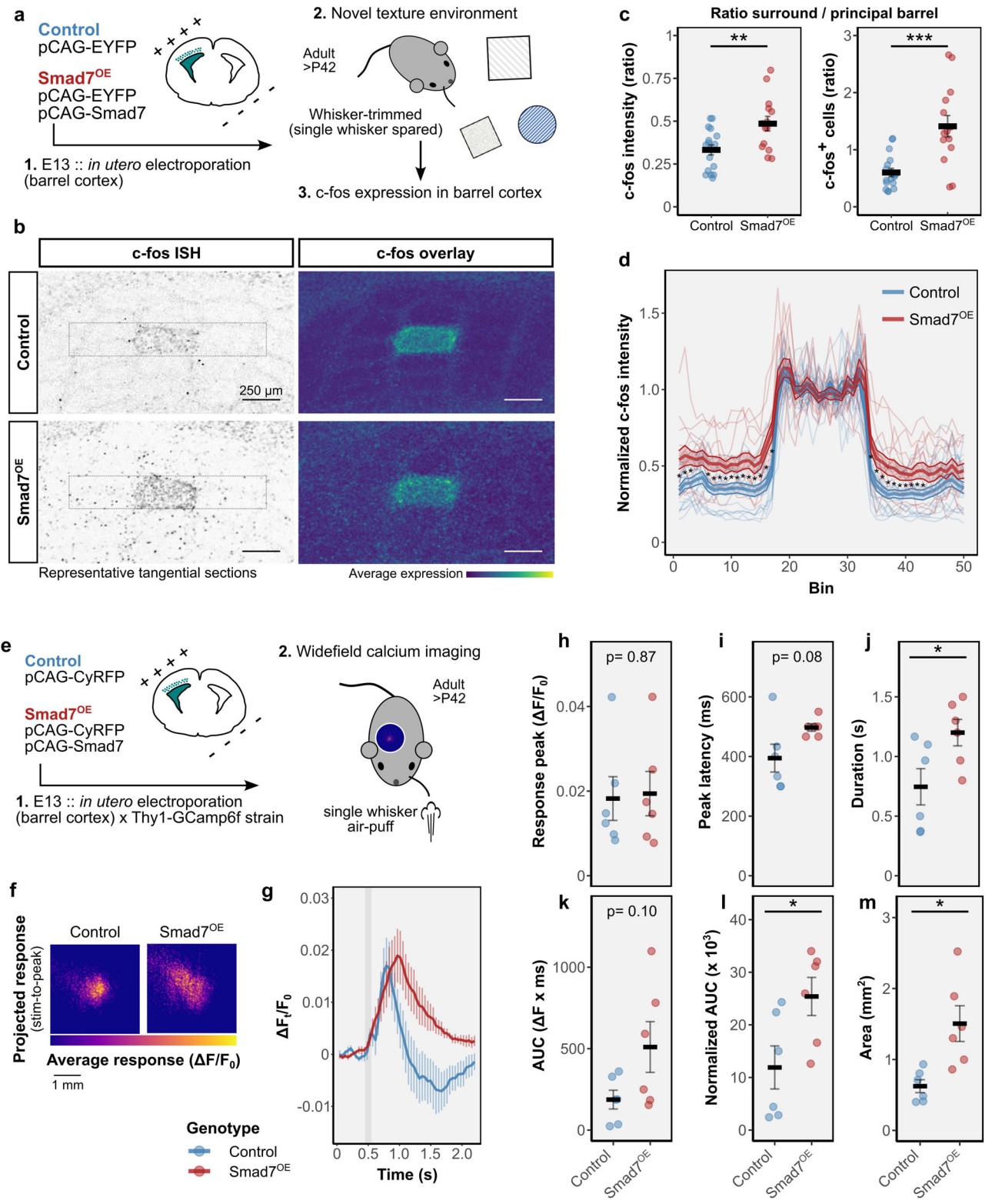

these animals[35]. Indeed, TCA inputs in Sert[Cre]; TeNT[Tg] mutants do not cluster appropriately in barrel cortex (Supplementary Fig. 3f, g), thus raise the possibility that activity acting upstream of TCA clustering is a critical step for providing appropriate barrel formation cues. Novel tools to manipulate TCA activity more precisely will be important here to elucidating activity-dependent mechanisms.

Notably, the altered gene expression and morphology that results from ectopic Smad7 expression raises the possibility of certain septa-

related genes being able to repress TCA-induced spiny stellate specification. While detailed mechanisms are needed, we have found a possible novel role for a form of TGF-β signaling involving the *Tgfbr1* receptor within the post-mitotic cortex, with expression of putative ligand (*Tgfb1, Tgfb3, Gdf10,* and/or *Inhbb*) and receptor (*Tgfbr1, Tgfbr3, Acvr1b*) interactions found in the VPM-barrel cortex pathway (Fig. 3 and Supplementary Fig. 4). Given the Tgfbr1 shRNA knockdown phenotype was not as severe as Smad7[OE] and the lack of barrel phenotype

**Fig. 6 | Functional consequences of spiny stellate-depleted barrel cortex assessed in Smad7[OE] mice. a** Schematic for assessing whisker-dependent neuronal activity in Smad7[OE] mice. A single whisker was spared in adult mice, which were then placed in a novel texture environment. Whisker-evoked activity in barrel cortex was assessed by c-fos expression. **b** Representative images for *fos* ISH in tangential sections through L4 barrel cortex (left). A heatmap of average expression for all animals is shown (right). Scale bars, 250 μm. **c** Ratio of *fos* expression intensity (left) and *fos*+ cells (right) were calculated in surrounding versus principal barrel (mean ± SEM). Asterisks indicate: **$p = 0.007$ and ***$p = 0.0009$, two-sided Welch's *t* test ($n = 17$ control and $n = 14$ Smad7[OE] brains). **d** Profiles of *fos* expression intensity, normalized to the peak barrel intensity (mean ± SEM), measured across the areas outlined in (**b**). Asterisk indicates $p < 0.05$, two-sided Welch's *t* test with multiple comparison correction by the Benjamini–Hochberg method ($n = 17$ control and

$n = 14$ Smad7[OE] brains). **e**, Schematic of calcium imaging in Thy1-GCaMP6f x Smad7[OE] mice. **f** Average response in barrel cortex upon whisker-stimulation. For each animal, a projection image was made using the averaged (over 60–90 trials) $\Delta F/F_0$ time-series during the post-stimulus response period (stimulus-to-peak). Shown is the average projection from control (left, $n = 6$) and Smad7[OE] (right, $n = 6$) animals. **g** Time course of GCaMP6f activity (mean ± SEM) in barrel cortex following single-whisker stimulation. Gray vertical line indicates whisker-stimulation. **h–m** Quantification of GCaMP6f activity (mean ± SEM) assessing **h** peak amplitude, **i** peak latency, **j** response duration, *$p = 0.039$, **k** area under curve (AUC), **l** normalized AUC (to peak amplitude), *$p = 0.033$ and **m** cortical area, *$p = 0.015$. In (**g–m**), displayed are group means ± SEM. Asterisk indicates *$p < 0.05$, two-sided Welch's *t* test, $n = 6$ control and $n = 6$ Smad7[OE] brains. Source data are provided as a Source data file.

in Smad7 conditional KO mice, it is possible that a non-canonical form of signaling operates in the setting of barrel formation. Whether ligand/receptor members other than *Tgfbr1* are also important will be critical in determining the precise signaling mechanisms of the pathway. One possibility is that Smad7 is acting upstream as a "brake" to maintain pyramidal fate, whereupon additional TCA-specific cues are required to both overcome (through repression of Smad7 expression) and initiate spiny stellate transcriptional events.

It has been reported that "higher-order" somatosensory cortical gene expression expands with the specific absence of VPM input, which is considered due to plasticity of higher-order thalamic nuclei[19]. We found, however, in the complete absence of thalamic innervation in the Celsr3 cKO[53], a similar expansion for the broadly expressed septa-related genes, suggesting a developmental similarity between pyramidal neurons in the septa and higher-order cortical areas. Thus, we hypothesize that the later-occurring post-mitotic emergence of L4 spiny stellate neurons represent a specialized cell-type for processing highly salient information regarding the external environment. Here, we note ethological differences in sensory modalities that align with distinct L4 spiny stellate neuron distributions observed across different species: for example, nocturnal rodent species display a high abundance of L4 spiny stellate neurons in whisker-related somatosensory cortex, with relatively few being found in visual cortex[68–70]. Conversely, spiny stellate neurons feature prominently in the visual cortex of binocular species (e.g., cats, ferrets, primates), compared to other sensory areas, and display a possible association with high acuity representations[17,71–74].

In this study, we demonstrated that the concomitant increase in L4 pyramidal neurons and loss of spiny stellate neurons induced by Smad7 overexpression led to more widespread connectivity and activation of the barrel cortex (Supplementary Fig. 11). Thus, it seems the ability of L4 to receive and relay discrete whisker-evoked information within discrete barrel columns is a functional role conferred by spiny stellate neurons (via TCA-orientated dendritic fields and dense local arborizations) and required for whisker-dependent discriminatory ability. It is possible that in these Smad7[OE]-disrupted circuits, an imbalance of discrete whisker-evoked sensory activity ("barrel" circuitry) and multi-whisker, sensorimotor signals ("septa" circuitry) further impairs fine spatial/texture discrimination. This likely arises from a maintenance of pyramidal morphology in Smad7[OE] neurons, thereby receiving and sending more distributed signals and adopting a more integrative role in the cortex. Indeed, Smad7[OE] neurons received more local inputs arising from outside their home barrel, similar to L4 pyramidal neurons found in barrel cortex[10]. However, the precise targets of the ectopic L4 contralateral projection and whether additional changes to local and long-range connectivity for Smad7[OE] neurons exists between other cell types (e.g., L4 SOM+ interneurons[70]) and/or cortical targets (e.g., ipsilateral M1 and S2[75,76]), was not explored in this study.

Overall, our findings highlight roles for spiny stellate neurons as sensitive coincidence detectors of sensory input to the cortex, helping

to amplify and conduct the initial flow of information to within functional cortical units[6,50,77]. Here, their critical position within barrel circuits is determined by specific TCA-derived cues (molecular and/or activity), that appear to drive a transcriptional program leading to spiny stellate specification from an immature cortical landscape.

## Methods
### Animals
Animals were housed at constant temperature (24 ± 2 °C) and relative humidity (55 ± 10%). All procedures were performed in accordance with protocols approved by the Institutional Animal Care and Use Committees of RIKEN Wako branch (W2022-2-026, 2022-035(1), 2022-027(1), W2019-131215), RIKEN Kobe branch (A2001-03), and the National Institute for Physiological Sciences (17A018, 18A032, 19A050, 20A034, 21A063, 22A018). All experiments conformed to RIKEN regulations for scientific research. Timed-pregnant ICR (CD-1) mice were sourced from Japan SLC. The day of discovery for the vaginal plug was considered as embryonic day (E) 0.5. Celsr3 conditional knockout (cKO) mice were obtained by crossing female Celsr3[f/f] mice[31,53] (RRID:MGI:3579160) with male Dlx5/6[IRES-Cre+/-]; Celsr3[f/+] mice[78] (RRID:MGI:3795743). Male Sert[Cre] mice (B6.129(Cg)-Slc6a4[tm1.1(cre)Xz/J], RRID:IMSR_JAX:014554)[79] were bred with female mice from a Cre-inducible tetanus toxin (TeNT) strain (B6;129S6-Gt(ROSA)26Sor[tm1.1(CAG-EGFP/TeNT)Imayo], RRID:IMSR_RBRC05154)[36] to obtain heterozygote TeNT[Tg/–] animals, with Cre-positive animals compared to Cre-negative controls. Fixed Sert[Cre]; Rim1[f/f]Rim2[f/f] (Rim1/2dKO, RRID:MGI:4939901) brain samples (age: P8) were obtained from Dr Patricia Gaspar[35]. For widefield calcium imaging, Thy1-GCaMP6f (RRID: IMSR_JAX:024276) male mice were mated with female C57BL/6 mice and in utero electroporation (IUE) was performed on pregnant dams at E13.5. R26-CAG-LSL-Smad7-HA (conditional Smad7-Tg) mice (Accession No. CDB0102E: https://large.riken.jp/distribution/mutant-list.html) were generated as previously described[80]. Briefly, Smad7-HA was cloned into a targeting vector containing homology arms (0.9 kb and 0.6 kb) for the Rosa26 locus, being inserted directly downstream of a pCAG-LoxP-STOP-LoxP (LSL) cassette[81,82]. Using Crispr/Cas9-mediated targeting (gRNA site: CGCCCATCTTCTAGAAAGAC), pCAG-LSL-Smad7-HA-polyA was then inserted into the Rosa26 locus by homologous recombination. Offspring were genotyped using the following primers: Fwd, (5'-GGG GGA GGA TTG GGA AGA CAA TAG C-3') and Rev, (5'-AGA ACT GCA GTG TTG AGG-3') (715 bp). F1 mice (C57BL/6 N background) were bred with Nex[Cre] mice for cortical overexpression of the Smad7 transgene.

All animals were housed under a 12 h light/dark cycle with water provided ad libitum. Sex was not considered as a biological variable in this study. Animals of either sex were used in experiments at postnatal (P0–P21) and/or adult (>P42) developmental ages. Recovery surgeries were performed either under isoflurane inhalation or with intraperitoneal injection of a mixture of medetomidine, midazolam, and butorphanol (MMB; 0.75/4/5 mg/kg), which was followed by atipamezole (0.75 mg/kg) for the reversal of anesthetic effects.

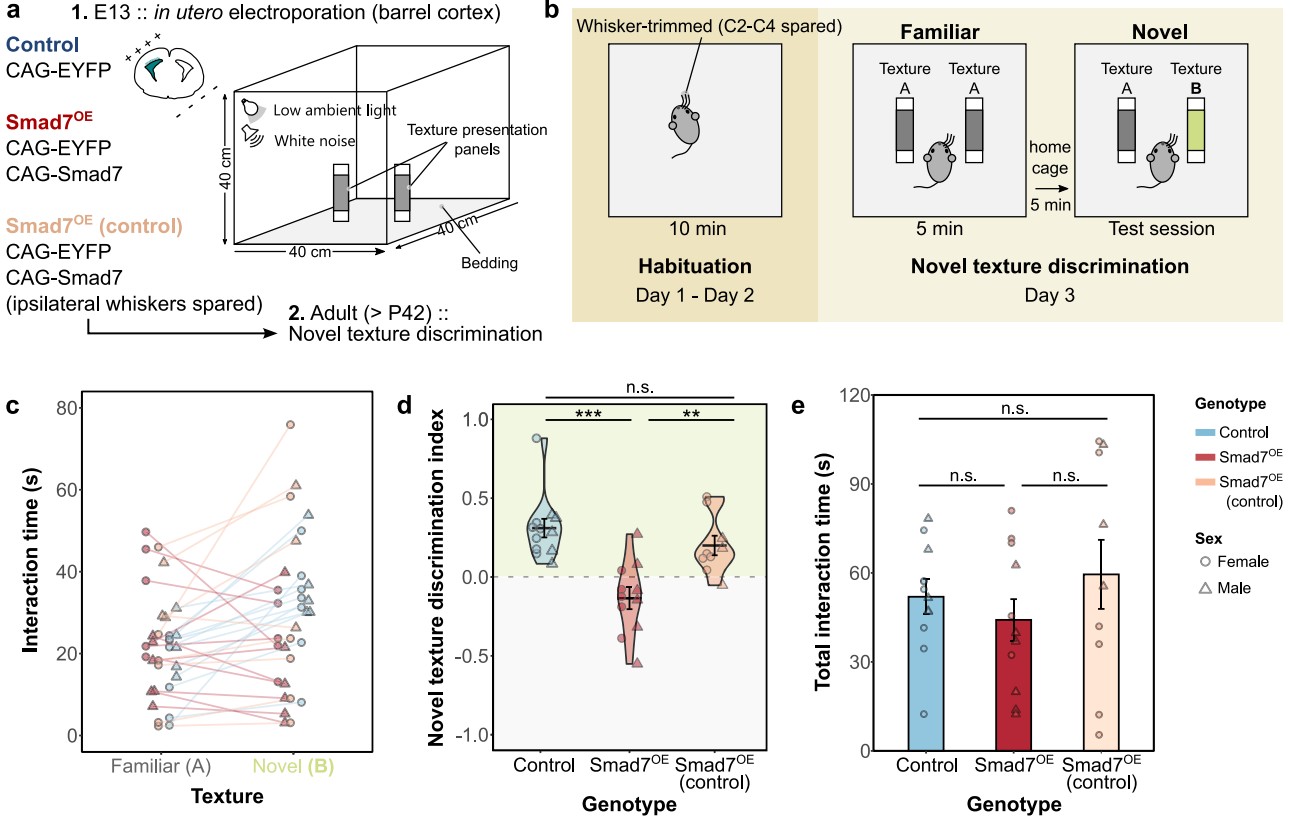

**Fig. 7 | Impaired ability of Smad7[OE] mice to discriminate during a whisker-dependent behavioral task. a** Smad7 overexpression was performed by IUE to target L4 barrel cortex, and whisker-trimmed, adult mice were assessed on the novel texture discrimination task (NTD). **b** Schematic for testing whisker-dependent texture discrimination. After habituation to the arena and presentation of similar textures ("A," gray), a novel texture ("B," green) is presented in the test session. **c** Interaction times for the familiar and novel texture during the test session for individual subjects in each group. **d** NTD index was calculated as follows: (interaction times for novel − familiar textures) divided by total interaction times.

No significant interaction between sex and genotype was found (ANOVA: $F_{2,26} = 0.28$, $p = 0.76$). Performance for each group is displayed as violin plots with group means ± SEM. A positive NTD ratio was observed for both control groups, indicating a preference towards the novel texture. This was not seen in the Smad7[OE] group. **e** Total interaction times were not significantly different between Smad7[OE] and control groups. Displayed are group means ± SEM. For (**d**, **e**), asterisks indicate **$p = 0.005$, ***$p = 0.0001$, ANOVA with Tukey's HSD test ($n = 7$ female and 5 male EYFP[OE] $n = 5$ female and 6 male Smad7[OE] $n = 6$ female and 3 male Smad7[OE-control] mice). Source data are provided as a Source data file.

## Sample collection

For collection of brain samples, animals were injected with a lethal dose of pentobarbital (≥100 mg/kg). After three failed attempts to elicit a foot withdrawal reflex, animals were perfused transcardially with 4% (w/v) paraformaldehyde (PFA) in phosphate buffer (PB). For adult mice, perfusion was first performed with 15 mL phosphate buffered saline (PBS) prior to PFA.

## Spatial transcriptomics using the 10× Visium platform

After a brief perfusion with RNase-free PBS, unfixed brains (age: P6–P7) were extracted, immediately embedded in Optimal Cutting Temperature (OCT) compound (Sakura Finetek), and frozen in a dry-ice cooled isopentane bath. Sections were made at 10 µm using a cryostat (Model HM525NX, ThermoFisher Scientific) and mounted on to the capture area of Visium Gene Expression slides (10× Genomics). Multiple sections containing the barrel cortex were collected from wildtype (C57BL/6 and ICR backgrounds) and from Celsr3[flox/flox] control and Dlx5/6[IRES-Cre]; Celsr3[flox/flox] mice. Multiple sections (10–15) from the same block were collected for assessment of RNA integrity. Mounted slides were stored at -80 °C until processing for spatial transcriptome profiling (Spatial 3′ v1 chemistry, 10x Genomics). Methanol fixation, tissue staining, permeabilization (18 min), and library preparation was performed according to the Visium Spatial Gene Expression User Guide (CG000239, 10x Genomics). Sequencing was performed on a NovaSeq

6000 System (Illumina) at a sequencing depth of approximately 110–400 × 10⁶ read-pairs per sample. The follow sequencing protocol was used: Read 1 (Spatial barcode): 28 cycles; i7 (sample) index read: 10 cycles; i5 (sample) index read: 10 cycles; Read 2 (Insert): 90 cycles.

**Visium 10x spatial transcriptomics analysis.** Following a 10x Space-Ranger software (v1.3.1) pipeline for genomic (reference genome: mm10-2020-A) and histological alignment of spatial sequencing data (raw FASTQ files), each spatial transcriptome (QC metrics in Supplementary Fig. 9) was filtered, normalized and log transformed using the Python scanpy (v1.9.1) toolkit (minimum gene counts = 200, minimum genes per cell = 10). Barcodes situated within somatosensory cortex were manually selected to obtain a region of interest (ROI) for downstream analyses. An annotated reference single-cell RNA-seq (scRNA-seq) dataset from P7 mouse somatosensory cortex (GSE204759) was used for instances where Tangram[32] was employed for scRNA integration with the spatial data. Using annotated cell-types in the pre-processed scRNA dataset, Tangram (mode = "clusters," density = "rna_count_based," num_epochs = 500) was used to predict the location of "Layer IV" barcodes in spatial data ROIs, using a $p = 0.6$ selection threshold. Prediction of scRNA transcript distribution on control sections was also performed using Tangram (mode = "cells").

Cluster analysis of L4 barcodes was performed in R using the Seurat package[83] (version 4.1). Here, L4 barcodes were selected using

the Tangram method above and merged into a single Seurat object. Following normalization and scaling of the data (SCTransform), clustering was performed with Harmony-implemented batch correction (v1.0) to account for section-to-section variability[84]. Here, a list of ISH-screened hollow- and septa-related genes served as the input for semi-supervised clustering of the L4 spots (see Supplementary Data 1). Gene signature scores for hollow and septa were calculated using the "AddModuleScore" Seurat function with hollow and septa gene lists as input (number of control genes = 400).

For differential expression analysis, raw counts from selected spots were analyzed in R using the DESeq2 package (v1.34.0)[85]. Section-to-section effects were included in the model design and normalization between conditions was performed using housekeeping gene, *Rpl19*. Statistical testing was performed using the Wald test, with Benjamini-Hochberg correction.

### In situ hybridization (ISH)

ISH was performed using previously described methods[86,87]. Here, key details of the ISH protocol are summarized in brief. For tangential sections, cortices were dissected after post-fixation then flattened under a petri dish (Ø = 6 cm) while submerged in 30% sucrose/PFA. After cryoprotection, brains were sectioned coronally at 28 μm, or tangentially at 40 μm, using a freezing sliding microtome (Leica). Plasmids used as cDNA templates for riboprobe synthesis were generated by PCR cloning or obtained from either the RIKEN FANTOM or IMAGE consortiums (Supplementary Data 3). Digoxigenin (DIG)-labeled riboprobes were synthesized by in vitro transcription (3 hrs at 37 °C) from 1 μg linearized DNA template following the manufacturer's instructions (Roche). Briefly, sections mounted/dried on Superfrost Plus slides (Fischer Scientific) were fixed twice with 4% PFA, before and after digestion with proteinase K (Roche). Hybridization with DIG-labeled riboprobes was performed overnight at 72 °C. Stringency washes were performed with 50% (v/v) formamide, 1% (w/v) SDS in 2× SSC. After blocking with sheep serum (Biowest), sections were incubated with anti-DIG antibody conjugated to horseradish peroxidase (1 in 5000 dilution, Roche). For signal development, a chromogenic color reaction was performed at room temperature using nitro-blue tetrazolium and 5-bromo, 4-chloro, 3-indolylphosphate (Nacalai, Japan).

### In utero electroporation (IUE)

Technical details of IUE have been described previously[88]. Here, we summarize key details for IUE of purified plasmid DNA. Electroporation was performed using a square wave electroporator (CUY21EDIT, Nepagene) at E13.5 (labeled as E13 in figures) to target cortical L4 neurons in primary somatosensory cortex. Five square wave pulses (50 ms duration, period 1 s) at 30 V were applied to embryos using round (3 mm) platinum plate tweezer electrodes (Nepagene). After electroporation, the abdominal cavity was sutured, and skin closed with wound clips (AutoClip, Fine Science Tools). All plasmids were used at 1 μg/μL, unless indicated otherwise. Sparse-labeling experiments were performed using either pCAG-LoxP-Neomycin-(**S**TOP)-**L**oxP (LSL)-EYFP or pCAG-LSL-DsRed2, along with a titrated amount of pCAG-Cre (0.2–1 ng/μL) to allow visualization of distinct neuron morphologies. Sparse **B**FP-[**H**istone- GFP]-**T**VA66T-[N2c **G**lycoprotein] (BHTG) rabies construct labeling was also achieved with a titrated amount of pCAG-Cre (1–20 ng/μL).

### DNA constructs

Plasmids for electroporation (pCAG-EYFP, pCAG-LSL-EYFP, pCAG-LSL-DsRed2, pCAG-Cre, pCAG3-MCS-HA) were generous gifts from Dr. Nobuhiko Yamamoto. The EGFP expression plasmid was obtained from Dr. Elizabeth Grove. pCAG-CyRFP1 was a gift from Dr. Ryohei Yasuda (Addgene plasmid #84356; RRID: Addgene_84356). The construct used for rabies virus tracing, pEf1a-DIO-BHTG, was a generous gift from Dr. Franck Polleux. RabV-CVS-N2cΔG-mCherry was a gift from Dr. Thomas Jessell (Addgene plasmid #73464; RRID: Addgene_73464). cDNA encoding mouse Smad7 was cloned into a pGEM vector using PCR primers: mSmad7-Fwd-XhoI, CTCGAGCCACCATGTTCAGGACCAAACGAT and mSmad7-Rev-NotI, GCGGCCGCCCGGCTGTTGAAGATGACCT. Smad7 was then cloned into a pCAG-MCS-HA vector, using XhoI/NotI restriction sites, to generate pCAG-Smad7-HA. pCAG-Smad7-NLS was generated by inserting annealed SV40-NLS oligonucleotides (SV40-NLS-Fwd, GGCCGAGGGTGGAGGTCCACCTAAAAAG AAGCGGAAAGTGGGTGGAGG and SV40-NLS-Rev, GGCCCCTCCACCC ACTTTCCGCTTCTTTTTAGGTGGACCTCCACCCTC) into the NotI site of pCAG-Smad7-HA. Mutant Smad7 (pCAG-Smad7$^{K312E,K316E}$) was generated by site-directed mutagenesis of pCAG-Smad7-HA using the following primers: mSmad7Δ2E-Fwd, CAAGAGTCAGCTGGTACAGGA AGTGCGGAGCGAGATCG and mSmad7Δ2E-Rev, CGATCTCGCTCCG-CACTTCCTGTACCAGCTGACTCTTG. To generate a Cre-dependent Smad7 expression plasmid (pCAG-LSL-Smad7), the EYFP fragment from LSL-EYFP was replaced with Smad7-HA using XhoI/BglII restriction sites. Tgfbr1-targeting (NM_009370.3) shRNA constructs were generated by cloning hairpin oligonucleotides with complementary overhangs into the BglII/HindIII site of a pSUPER.retro.neo vector, according to the manufacturer's guidelines (oligoengine). The following target sequences were designed based on existing guidelines[88–91] using the Eurofins siRNA design tool (sense sequence): Tgfbr1-sHRNA-#1, AGAATATCTTGGTGAAGAA; Tgfbr1-sHRNA-#2, GAT ATTCTCTTCTAGAGAA; Tgfbr1-sHRNA-#3, GTTCTAGATGATTCCA-TAA. Constructs were used individually (Tgfbr1-sHRNA-#1) or in combination at equimolar concentrations and in both cases confirmed to reduce expression in transfected cortex using a *Tgfbr1* riboprobe.

### Immunohistochemistry

Following perfusion, brains were dissected, post-fixed in 4% PFA for 1 h at 4 °C, then cryoprotected in 30% (w/v) sucrose dissolved in 0.1 M PBS. After cryoprotection, brains were sectioned coronally at 28–60 μm, using a freezing sledge microtome (Leica), and mounted/dried onto Superfrost Plus slides (Fisher Scientific). Antibody dilution and blocking was performed using 5% (v/v) normal goat serum, 0.2% (v/v) Triton-X/PBS. Briefly, after blocking overnight at 4 °C, immunostaining with primary antibodies was performed at 4 °C for 24–48 h. Slides were then washed 3 times for 1 h in 0.5% Triton-X/PBS. Labeling with appropriate secondary antibodies (goat anti-rabbit IgG (H + L) Highly Cross-Adsorbed Secondary Antibody, Alexa Fluor 488, RRID:AB_2576217, Thermo Fisher Scientific; goat anti-rabbit IgG (H + L) Highly Cross-Adsorbed Secondary Antibody, Alexa Fluor 594, RRID:AB_2534095, Thermo Fisher Scientific; goat anti-rat IgG (H + L) Cross-Adsorbed Secondary Antibody, Alexa Fluor 488, RRID:AB_2534074, Thermo Fisher Scientific; goat anti-rat IgG (H + L) Cross-Adsorbed Secondary Antibody, Alexa Fluor 647, RRID:AB_141778, Thermo Fisher Scientific; Cy3 AffiniPure donkey anti-guinea Pig IgG (H + L), RRID: AB_2340460, Jackson ImmunoReseaerch; Alexa Fluor 647 AffiniPure donkey anti-guinea Pig IgG (H + L), RRID: AB_2340476, Jackson ImmunoResearch) was performed overnight at 4 °C (1 in 500 dilution). After a 30 min wash in PBS, sections were counterstained with DAPI, followed by 3 × 30 min washes in PBS. Slides were coverslipped using Mowoil/DABCO mounting medium. Primary antibodies and dilution factors used in this study were rabbit anti-GFP (1 in 500; RRID:AB_591819, MBL International), rat anti-GFP (1 in 500; RRID:AB_10013361, Nacalai), rabbit anti-DsRed (1 in 500; RRID:AB_10013483, Takara Bio), rat anti-HA (1 in 500; RRID:AB_390919, Roche), rabbit anti-CTB (1 in 500: RRID:AB_726859, Abcam), guinea pig anti-vGlut2 (1 in 500; RRID:AB_887884, Synaptic Systems), mouse anti-RORbeta (1 in 100; RRID:AB_1964364, Cosmo Bio), mouse anti-ROR common (1 in 100; RRID:AB_605116, Cosmo Bio), rabbit anti-Smad7 (1 in 100; RRID: AB_2889839, Abcam). For some antigens (RORc, RORbeta, Smad7), antigen retrieval (10 min at 90 °C in 10 mM citrate buffer, pH 7.0) was performed prior to immunostaining.

Cytochrome oxidase staining was performed as previously described[92]. Briefly, mounted sections were incubated overnight at 37 °C in staining solution containing ~0.01% (w/v) cytochrome c (Nacalai) and 0.02% (w/v) 3′,3′-diaminobenzidine (DAB) (Sigma-Aldrich) dissolved in PB.

## Histology image analysis

Fluorescent images were acquired using a Keyence BZ-X700 microscope with appropriate filter sets. Cell morphology was assessed using either an FV-1000 or FV-3000 (Olympus) confocal microscope. Acquired data were imported and analyzed using ImageJ (FIJI, v1.53j). A control dataset from Celsr3 controls was used for comparisons to both Celsr3cKO and Smad7[OE], sharing a common ICR genetic background. For categorization of barrel cell-types, we used criteria described previously[26]. Spiny stellate neurons lacked an apical dendrite that did not extend beyond L4, while often displayed asymmetry in their dendrites toward TCA clusters in barrel centers. Pyramidal cells (star pyramidal and pyramidal cells were grouped) had a prominent apical dendrite which often extended to L1 where it made a number of tuft-like branches. For non-fluorescent images, slides were scanned under brightfield illumination using a SCN400 slide scanner (Leica) or Nanozoomer automated slide scanner (Hamamatsu). For analysis of gene expression in Celsr3 cKOs, ISH images were converted to inverted, greyscale images and background subtracted (rolling ball radius=300). Distribution of signal was determined from an intensity profile across the cortical layers using a region-of-interest (ROI) that was consistent for all sample images. Profiles were binned into 10 equal regions and differences in Celsr3 cKO expression profiles were calculated relative to control samples, expressed as log2FoldChange. For each gene, intensity profiles were averaged over sections from at least two brains per genotype. Genes used in analysis: *Astn2, Btbd3, Cadps2, Nr1d1, Ntng2, Dapk2, Rasd2, Barx2, Ier5, Map1b, Nrxn2, Sez6l2, Sorl1, Syn2, Rac3, Met, Tppp3, Cd24a*. Cell counting was performed using CellProfiler[93] (v4.2.5)

For comparison of expression in Smad7[OE] brains, sections stained with GFP riboprobe were used to firstly assess the region of in utero electroporation (IUE). Brains in which the IUE region did not target barrel cortex were not included in downstream analyses. This was determined by overlaying the GFP image with corresponding serial sections hybridized with *Rorb* probe, which shows broader, clustered expression in L4 barrel cortex. In some brains, the position of unaltered barrel marker expression in the un-electroporated cortex was used as a proxy. A corresponding ROI in the un-electroporated hemisphere was chosen based on the position of the IUE ROI. By using the GFP image overlaid with corresponding serial sections that were hybridized with genes of interest, ROIs should be consistent across sections (although not accounting for slight differences in the anterior-posterior position of serial sections, offset by 28–56 μm). For analysis of expression, a plot profile was generated (ImageJ) and expression intensity was binned along the radial axis of the cortex (100 bins). Log2FoldChange (IUE+ vs IUE− side) was calculated in L4.

## Retrograde CTB labeling of contralateral projections

Cholera-toxin subunit B (CTB) conjugated to AlexaFluor594 (Invitrogen) was prepared at 0.5% (w/v), dissolved in PBS and stored at −20 °C. Stereotaxic injections were made into the contralateral corpus callosum of adult mice transfected by IUE. A total volume of 650 nL was infused at a rate 125 nL/min. Coordinates used were A-P, 1.4 mm; M-L, 0.9 mm; D-V, 1.4 mm, angle = 8°. At least 48 hrs post injection, animals were sacrificed and brains were collected as above for immunohistochemistry. Detection of CTB signal was enhanced by immunostaining of coronal sections (50 μm), as described above.

## Slice electrophysiology

Oblique coronal slices of the barrel cortex (300 μm thick) were prepared from P13-P19 mice (naïve ICR and Smad7[OE]), isolated under deep anesthesia with isoflurane. Slices were recovered for 1 h in oxygenated (95% $O_2$ and 5% $CO_2$) normal artificial cerebrospinal fluid (ACSF) containing (in mM): 126 NaCl, 3 KCl, 1.3 $MgSO_4$, 2.4 $CaCl_2$, 1.2 $NaH_2PO_4$, 26 $NaHCO_3$, and 10 glucose at 33 °C. After recovery, slices were kept in ACSF at 25 °C. For whole-cell recording, patch pipettes (4−6 MΩ) were filled with a solution containing (mM) 130 K-gluconate, 8 KCl, 1 $MgCl_2$, 0.6 EGTA, 10 HEPES, 3 Mg-ATP, 0.5 Na-GTP, 10 Na-phosphocreatine, and 0.2% biocytin (pH 7.3 with KOH). For analysis, we selected cells with a high seal resistance (>1 GΩ) and a series resistance <40 MΩ. During recordings, slices were perfused with modified ACSF containing 4 mM $CaCl_2$ and 4 mM $MgCl_2$ to reduce spontaneous EPSCs. Photostimulation was achieved by focal photolysis of Rubi-caged glutamate (150 μM) with 10 ms flashes of blue light (440 nm, 2.5 mW at the specimen plane) from a diode laser. The light was focused on the slices through a ×4, 0.16NA microscope objective. This resulted in the generation of action potentials in neurons with cell bodies situated within ~100 μm of the center of the illuminated spot (Supplementary Fig. 10). Photostimulation-evoked EPSCs were recorded from L4 neurons. Photostimulation was usually applied once every 5 s to each of the 15 × 28 sites in a quasi-random sequence. Maps of photostimulation sites were aligned to laminar borders, which were determined according to IR-DIC images captured during recording. We measured the peak amplitude of all EPSCs occurring within 150 ms after the stimulation and constructed color-coded maps using the total amplitude of the EPSCs at each stimulation site. The data were analyzed using Mini Analysis Program (Synaptosoft) and custom software written in Matlab. After recordings, slices were fixed by 4% PFA for biocytin staining with Streptavidin, Alexa Fluor™ 647 conjugate. When recorded cells were not characterized as spiny stellate cells in control mice, the data were not included in the analysis. The border of L4 barrels was confirmed with vGlut2 immunostaining. This setup was also used to determine L4 morphologies in Nex[Cre]; Smad7[Tg] barrel cortex at P7-P8.

## Rabies-mediated monosynaptic retrograde tracing

EnvA-pseudotyped, glycoprotein-deleted CVS-N2c rabies virus expressing mCherry (EnvA-CVS-N2c(ΔG)-RabV-mCherry) was produced in the Johansen lab according to previous methods[94]. Expression of BHTG was targeted to L4 barrel cortex by IUE at E13.5. Cre-dependent plasmids pCAG-LSL-Smad7 and/or pEF1a-DIO-[**B**FP-(**H**istone-GFP)-**T**VA66T-(N2c **G**lycoprotein)] (BHTG, control) were used with pCAG-Cre (1–20 ng/μL) to achieve sparse labeling. Stereotaxic surgery was performed in adult mice, targeting L4 barrel cortex (A–P: 1.2 mm; M–L: 3.5 mm; D-V: 0.4−0.5 mm relative to bregma and the pial surface). A total of 200 nL of EnvA-CVS-N2c(ΔG)-RabV-mCherry (5.77 × 10$^7$ U/mL) was injected using a 34 G needle at a rate of 40 nL/min. After 7−9 days, brains were collected as above for immunohistochemistry. Serial coronal sections (50 μm) were collected spanning most of the brain and immunostaining performed for GFP and mCherry.

**Rabies virus tracing image acquisition and analysis.** Sections were imaged as a *z*-stack using an automated stage fluorescence microscope (Keyence, BZ-X700) using a ×4 objective (NA 0.2; CFI Plan Apo λ, Nikon). Image stitching was performed using the FIJI Stitching plugin[95]. Sections were registered to the Allen Reference Atlas adult brain[96] (RRID:SCR_020999) using AMaSiNe software (v1.0)[97] with slight modifications. Preprocessing was performed using a Python implementation of the original Matlab code with the following modules: numpy (v1.18.1), opencv2 (v3.4.2.16), scikit-image (v0.16.2), scanpy (v1.9.1),

scipy (v1.5.2). Following an initial registration to the reference atlas, b-spline transformations were performed using SimpleITK (v1.2.4), with user-defined control points to more accurately align each section. Semi-automated counting of presynaptic (mCherry⁺) and starter neurons (GFP⁺/mCherry⁺) was manually curated and registered to the annotated brain regions used in the Allen Reference Atlas.

### Whisker-related c-fos activity in barrel cortex

Experiments involved EYFP$^{OE}$ and Smad7/EYFP$^{OE}$ animals transfected by IUE. Prior to testing, whiskers of adult mice were trimmed under light isoflurane anesthesia, sparing a single whisker (C2) contralateral to the cortex being examined. After recovery, mice were placed in a clean cage containing objects of different textures (coarse and fine sandpaper, cotton balls, aluminum foil, plastic and wooden dowels). After 2 hrs exploration within this novel texture environment, animals were sacrificed and brains collected. For tangential sections, brains were post-fixed in 4% PFA for 1 h at 4 °C, then flattened under two plastic petri dishes in 30% sucrose/4% PFA. After at least 2 days, 40 μm sections were made starting from the pial surface. In order to identify correctly-targeted electroporations, EYFP fluorescence was imaged immediately after mounting sections and used for subsequent alignment with brightfield images obtained after staining for c-fos expression. Sections were processed for *fos* ISH using the methods described above.

**Analysis of c-fos expression.** Sections containing EYFP signal overlapping with *fos* ISH signal within the C2 whisker region were used for analysis. Brightfield images containing *fos* expression were converted to greyscale, positioned (via rotation/translation) such that the C2-barrel was at the image center, then cropped to a constant dimension, containing neighboring barrels. Background was subtracted (rolling ball radius=50) and a plot profile of pixel intensities was measured across the B2-C2-D2 barrels. To account for differences in signal development, intensities were binned and normalized to the average intensity found within the C2 region. Heatmaps of overall c-fos signals were generated by averaging automatically thresholded images (ImageJ) for each group. Neurons positive for *fos* signal were also manually counted within the principal C2-barrel and its adjacent, neighboring barrels. A ratio of *fos*⁺ neurons outside vs inside the principal barrel was calculated.

### Widefield calcium imaging

**Surgery and image acquisition.** After IUE of pCAG-Smad7 and/or pCAG-CyRFP (control) plasmids into a Thy1-GCaMP6f$^{+/-}$ background, surgery was performed in adult mice to implant a cranial window consisting of a glass coverslip (Ø 5–6 mm) and supporting framework for widefield imaging. Briefly, after induction of gas anesthesia (1.5–2.5% isoflurane; Pfizer), injections of antibiotic (Baytril®; 2.5–5 mg/kg, subcutaneous; Bayer Yakuhin), anti-inflammatory (dexamethasone; 0.05−0.1 mL/kg, subcutaneous; Kyoritsu Seiyaku), anti-edema (Mannitol; 200 μL/20 g, intraperitoneal), and analgesic agents (Lepetan®; 0.2−0.3 mg/kg, subcutaneous; Otsuka Pharmaceuticals) were performed prior to removing the skin and periosteum covering the skull. A dental drill was used to perform a 5-mm-diameter craniotomy over barrel cortex of the electroporated hemisphere (centered 3.0 mm lateral and 0.9 mm posterior to bregma). A glass window consisting of an inner 5mm-diameter coverslip and outer 6-mm-diameter coverslip was positioned over the craniotomy and fixed to the skull using cyanoacrylic glue. A metal chamber and headpost was cemented to the surrounding skull with dental adhesive (Super-Bond C&B®, Sun Medical). At least two weeks following implant surgery, widefield calcium imaging was performed under light anesthesia (0.5% isoflurane, 0.2% air) in a head-restrained setup for widefield imaging[98,99]. Prior to imaging, all whiskers were trimmed apart from the C2 whisker, contralateral to the electroporated cortex. Hemodynamic and GCaMP6f fluorescent signals were

imaged using a sCMOS camera (PCO edge 5.5, PCO SDK (v1.14.0). In order to account for hemodynamic signals, most experiments (10 out of 12 animals) utilized interleaved dual-wavelength imaging with two shutter-controlled LEDs centered at 465 nm (LEX-2, Brainvision Inc.) and 405 nm (M405L2, Thorlabs). Synchronization of image capture (acquisition framerate: 60 Hz) and LED illumination was achieved using Arduino-controlled software. For initial experiments, single-wavelength (465 nm) illumination at an acquisition framerate of 40 Hz was used. In each trial, a pneumatic pump (PicoPump, PV830, World Precision Instruments) delivered a single air puff (10 ms pulse width, 10 PSI) through a glass capillary (Ø = 1 mm) positioned 1 cm in front of the whisker. For each trial, frames were acquired over a 2.5 s duration, with a pre-stimulus period of 500 ms. Trials were initiated every 10 s and at least 60 trials were performed for each imaging session. Image acquisition and whisker stimulation were initiated through a custom Matlab script.

**Image processing and analysis.** Images were spatially binned (4 × 4) to 220 × 200 pixels (each pixel representing ~30 μm × 30 μm). For each trial, capture frames were linearly detrended, followed by a high-pass (0.01 Hz) and low-pass (10 Hz) second-order butterworth filter. For hemodynamic correction, per-pixel averages were calculated for both 465 nm and 405 nm signals. A scaled 405 nm signal was obtained by multiplying the 405 nm signal by the 465 nm/405 nm ratio of per-pixel averages. The 465 nm signal was then subtracted from the scaled 405 nm to obtain the hemocorrected signal. $F_0$ was calculated as the average intensity during the baseline 500 ms pre-stimulus period. At each timepoint, $\Delta F_t$ was obtained with the following calculation: $\Delta F_t = (F_t - F_0)/F_0$. For each subject, a ROI of fixed size was manually selected over the barrel cortex displaying whisker-evoked activity (within 150 ms of the stimulus). A profile of pixel intensities within this region was calculated over the trial duration. The response peak was measured as the maximum $\Delta F/F_0$ following stimulus onset. The peak latency was calculated as the time from stimulus to response peak. Response duration was calculated as the time for the evoked response ($\Delta F$) to return to 20% of the peak response. Area under the curve (AUC) was determined by summing $\Delta F$ during the response duration, with normalization to the peak response. The area of the evoked response was determined by first performing a maximum projection of frames (stimulus-to-peak) centered on location of the peak response. Area was measured from the projected images after thresholding to 2 standard deviations of pixel intensity across the projection frame.

**Novel texture discrimination task.** To assess mice on a whisker-related behavior, mice were subjected to a novel texture discrimination task[52]. This task was based on the normal exploratory behavior of mice to investigate changes in their environment. Such novelty was introduced in the form of textures and was used to investigate whisker-based texture discrimination. Briefly, prior to testing, the whiskers of adult mice (P42 and older) were trimmed under brief (<5 min) isoflurane anesthesia, but unilaterally leaving three whiskers (C2-C4) intact on one side of the snout. For electroporated animals, the spared whiskers were contralateral to the transfected cortex. Animals for which whiskers ipsilateral to the electroporated cortex remained intact, served as an additional control group. Over 2 days, mice were habituated to an uncovered, white opaque Perspex arena (40 cm × 40 cm × 40 cm) for 10 min each day. On the third day, whiskers were re-trimmed, and testing was performed. Here, two vertical panels, displaying sandpaper of the same coarseness (texture A, P80 grade), were introduced into the center of the arena. After a 5 min session to explore the textures (A vs A), animals were returned to their home cage for 5 min. During this period, the two textures were replaced with another pair of panels displaying either a novel texture (texture B, P100

grade) or the previously encountered texture. The side of the novel texture was randomly assigned. Mice were returned to the arena and novel texture discrimination (A vs B) was assessed over a 5 min period. Habituation and testing were performed under low ambient lighting (<10 lux) and white noise. Videos were recorded from above the arena using a Webcam (Logitech) at 30fps under infrared illumination. A novel texture discrimination ratio was calculated as: (time spent interacting with the novel texture – time spent interacting with the original texture)/(total time interacting with both textures). Here, an "interaction" was measured as the time the animal spent within a 4 cm×4 cm region around the panel and in which a "whisker-directed" contact was made with the texture. Movements through the region without initiating such contact were not counted. The few occasions involving climbing onto the panels were also excluded.

**Statistics and reproducibility.** All micrographs were replicated in 3 or more independent brain samples. Statistical testing was performed in R. Normality was assessed using the Shapiro–Wilk test, whereas homogeneity of variance was assessed using Levene's test. Significance was reached for a test when $p < 0.05$. The details of statistical testing for each experiment are described in the manuscript text and figure legends.

**Reagents and resources.** Information regarding DNA constructs, reagents, and resources can be found in Supplementary Data 3.

### Reporting summary
Further information on research design is available in the Nature Portfolio Reporting Summary linked to this article.

## Data availability
The spatial transcriptomics data generated in this study have been deposited in the CBS repository system (https://doi.org/10.60178/cbs.20230816-001). Other data generated and/or analyzed in this study are available from the corresponding author upon reasonable request. This paper also analyzes existing, publicly available data: Allen Mouse Brain Adult Reference Atlas (CCFv3; https://atlas.brain-map.org/), scRNA-seq datasets (accession codes: GSM5277845, GSE204759). Source data are provided with this paper.

## Code availability
Original code generated in this study has been deposited in the CBS repository system (https://doi.org/10.60178/cbs.20230816-001).

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

## Acknowledgements

We thank Yayoi Nozaki, Mami U, Masae Sato, Kohei Onishi, Rie Nishiyama, and Javier Orlandi for technical assistance. We thank Dr. Andre Goffinet (Celsr3 cKO; Université Catholique de Louvain), Dr. Sandra Goebbels (NEX^Cre Max Planck Institute for Multidisciplinary Sciences), and Dr. Patricia Gaspar (Sert^Cre and Sert^Cre; Rim1^f/f Rim2;^f/f INSERM) for kindly providing transgenic mouse strains and samples. We also thank Dr. Franck Polleux (Columbia University) for providing the rabies virus plasmid. This work was supported by the Cooperative Study Program (17-116) of National Institute for Physiological Sciences (Y.Y., T.S.), the RIKEN Center for Brain Science (T.S.), RIKEN SPDR Fellowship (T.R.Y.), JSPS Fellowship (T.R.Y.; 25.03737), the Funding Program by MEXT Grants-in-Aid for Scientific Research on Innovative Areas "Cell type census of adaptive neuronal circuits: biological mechanisms of structural and functional organization (Adaptive Circuit Census, ACC)" (KAKENHI 21H05240) (T.S.).

## Author contributions

Conceptualization, T.R.Y. and T.S.; methodology, T.R.Y., A.B., J.J., Y.Y., and T.S.; investigation, T.R.Y. M.Y., S.S.K., and A.C.Y, generation of mouse strain, K.I. and T.A.; writing—original draft, T.R.Y.; writing—review and editing, T.R.Y., M.Y., Y.Y., and T.S.; funding acquisition, T.R.Y., Y.Y., and T.S.; resources, A.B., J.P.J., Y.Y., and T.S.; supervision, T.S.

## Competing interests

The authors declare no competing interests.
