## [Peer Review File · Nature Communications]

Thalamocortical control of cell-type specificity drives a circuit for processing whisker-related information in mouse barrel cortexREVIEWER COMMENTS

Reviewer #1 (Remarks to the Author):

This is a “tour de force” study where the authors used a wide array of cutting-edge techniques. The approach and the collected data so expansive that there is enough material easily for two PhD theses (at least by European standards). I congratulate the authors and the PI of this study for such an extensive and multifaceted approach to determine molecular and activity-dependent factors that specify spiny stellate barrel cells and star pyramid septal cells in layer 4 of the mouse primary somatosensory cortex.

The overall conclusion of the study is that the positioning and morphological specification of spiny stellate cells around whisker-related barrel walls is determined by specific thalamocortical axon(TCA)-derived cues that drive a transcriptional program leading to spiny stellate specification from an immature cortical landscape that is by default pyramidal (lines 493-498).

The authors start out from the premise that there is a lack of a detailed molecular characterization underlying the diversity of barrel hollow and septal neurons and L4 barrel cortical cell-types cannot be easily separated on the basis of their electrophysiological properties. They use a genome-wide spatial transcriptomics platform and identify complementary barrel and septa-related patterns of gene expression that have a close spatiotemporal relationship with thalamic innervation. Through various molecular and genetic manipulation approaches they find that a default cortical state is of septa/pyramidal identity. Specific TCA input patterns layer 4 into its barrel and septa divisions. Ectopic expression of a septa-related gene, *Smad7*, prevents L4 neurons from acquiring spiny stellate morphology, even with TCA innervation intact. Loss of stellate cell populations result in an increase of septa-like cellular morphologies and connectivity. As a consequence, there is widespread whisker evoked cortical activation, and an impaired ability to perform whisker-dependent texture discrimination.

Overall, the studies are well-designed, well-controlled, and appropriate statistics are used, even though sex is not considered as a biological variable, but that is probably not an issue in barrel cortex development.

The study spans molecular genetics, a genome-wide spatial transcriptomics to axonal/cellular labeling, in utero electroporation for cell type specific gene manipulation, activity-dependent gene expression patterns and density, slice electrophysiology, in vivo calcium imaging, whisker sensory discrimination behavioral tests. Overall, the study is well designed, with appropriate rigor and statistics

The idea and conclusion are quite novel: Cortical neurons are pyramidal by default but interactions with TCAs through genetic, molecular, activity-dependent processes are specified into spiny stellate cells that form the barrel walls. They first search for genetic markers that can be used to distinguish between the layer 4 barrel walls and barrel septa. They used a genome-wide spatial transcriptomics platform, and identified reciprocal barrel- and septa-related patterns of gene expression that was in close register with the timing of TCA innervation and barrel formation. Their evidence of a default cortical state that is of septa/pyramidal identity and TCA inputs pattern the barrel cortex Layer 4 into its barrel and septa divisions. Ectopic expression of a septa-related gene, *Smad7*, prevents morphological transformations into spiny stellate cells, even with intact TCA innervation. Loss of stellate cell populations result in an increase of septa-like connectivity (mostly callosal), widespread whisker evoked cortical activation, and an impaired ability to perform whisker-dependent texture discrimination. However, these conclusions may be a bit premature and should include further considerations as detailed before.

The main issues I had concerns over are with the figures and also with one of the main mouse models used in the study.

1. The methods (lines 825-26) note that “*Celsr3* conditional knockout (cKO) mice were obtained by crossing female *Celsr3*^{f/f} mice^{31,54} (RRID:MGI:3579160) with male *Celsr3*^{f/+};*Dlx5/6*-Cre^{+/-} mice⁶⁵ (RRID:MGI:3795743)”. And the authors note in lines 111-113 that they used *Dlx5/6*^{cre};*Celsr3*^{flox/flox} (*Celsr3* conditional knockout) mice which lack TCA innervation of the cortex and as a result, fail to form barrels. There is no further information given about this mouse line which is a concern. Because there are published reports showing that *Celsr3* gene is required for interneuron migration in the mouse forebrain (PMID: 19332558); *Celsr3* gene is involved in globus pallidus development and connectivity (PMID: 25113559); *Celsr3* inactivation in the telencephalon, in the ventral forebrain, or in the cortex have essential roles in both subcerebral projections (including the spinal cord) of cortical neurons and , thalamocortical axons (TCAs) that project through the ventral forebrain (PMID: 19349379) and along with *FGzd3* regulate reciprocal thalamocortical projections independently of their expression in cortical or thalamic neurons. (PMID: 27170656). *Celsr3* is also involved in interneuron migration in the mouse brain (PMID 19332558). *Celsr3* ko mice exhibit loss of 50% of glutamatergic synapses in pyramidal neurons of the hippocampus (PMID 28057866). These multitude of *Celsr3* functions during brain development deserve some discussion at least.

Likewise, *Dlx5/6* are expressed by developing and mature GABAergic interneurons and play a role in the differentiation of certain GABAergic subclasses (PMID: 31514171, PMID: 35681437).

There are several important issues that need to be clarified and discussed. Since the *Celsr3* cKO cortex does not get TCAs, one would expect some morphological changes in layer 4 but from the extended Figure 1 it does not seem so, and that the cortical thickness looks remarkably similar. Which is very peculiar if the dense TCA innervation in layer 4 is lacking what makes its thickness equivalent to controls? Does layer 6 still project back to the VPM even when VPM does not project to the S1 cortex

(see references in above paragraph about disrupted reciprocal connections following Celsr3 mutation) ? Is this unique to the VPM-S1 circuitry or generalized across thalamic sensory nuclei and their cortical primary sensory targets? Figure 1 indicates that the Celsr3 cKO VPM projects down to and towards the limbic cortex and the cerebral peduncle, where do they end up? Does this mutation also affect VPM/VPL size?

Figures:

It would be much better if the authors would attend to consistency in scale bars they use in the figures. Even in the same group of micrographs in a sub figure they have one scale bar labeled 100 um and several other differing size scale bars with no labels, are they all 100 um?

Several of the photomicrographs in the figures appear as if they are prepared by a miniaturist; difficult to discern what they are supposed to show (even when the pdf is magnified 400x).

Figure 1: Figure 1b is a drawing, it would be better to show actual photomicrographs of sections, for example, Nissl stained to see the absence of the barrels in Celsr3 cKO cortex.

In Figure 1c micrographs are too dark to assess them. It is not clear why Celsr3 cKO cortex is stained for RORc (also no green staining is visible in the presented micrographs) and the control cortex vGlut2. Presumably due to absence of TCAs in the ko cortex but even so, vGlut2 staining for both genotypes would be a better pair. With the exception of top right micrograph from the ko cases the other "pyramidal" cells do not look convincingly pyramidal, in fact they look like cells with retarded dendritic development. In these micrographs the authors should use the same scale bar, if they are all 100 um, then the left top panel ko sample the cell bodies would be at least twice the size of other pyramidal cells.

Figure 1h also needs scale bar. I am puzzled by the morphological appearance of the pyramidal neurons in top right frame, they amazingly look like layer 5 pyramids, just underneath layer 4, they have far more apical branches as they approach the pial surface, and also distinct basal dendrites compared to their control partners, which clearly look like star pyramids. Could there be a mix up?

Is it possible to show c-fos activity in the Celsr3 cKO cortex following whisker stimulation as a control to ensure that the whisker information isn't getting there?

Figure 2: vGlut2 staining photos are too dark especially at P2 there should be TCAs in presumptive layer 4 even when I magnify the paper 200 times, I can barely see them. In this figure too there is inconsistent use of scale bars.

Figure 2c is difficult to interpret, what is the explanation for DsRed2 labeled layer 3 cells especially at P2?

Figure 2c legend reads: "By P4, a clear distinction between spiny stellate (pink) and pyramidal morphologies (blue) could be made, with progressively more spiny stellate neurons found at the edge of barrels (visualized by vGlut2staining)" I can't see any blue cells with pyramidal morphology, all I can see is a few pink, immature looking pyramidal cells bodies

In Figure 2c. sparse labeling of layer 4 cells with DsRed2 by IUE at P13.5 has labeled cells in both layer 4 and above and a few blue arrowheads are pointing to black spaces. At P2, "shorter apical dendrites" can also be due to plane of section. "By P4, a clear distinction between spiny stellate (pink) and pyramidal morphologies (blue) could be made" in this picture there is only one pink cell with stellate morphology.

Figure 3: "Overexpression of septa-related gene, Smad7, recapitulates gene

expression and morphology changes seen in Celsr3 cKO barrel cortex". Figure 3b, Tgfb1 why is the VPM in this case notably shrunken? In all sections what is labeled as VPM is actually VPM+VPL. This figure also has inconsistent scale bar use. In d, how were the barrel outlines (dashed lines) determined?

This reviewer is a bit puzzled by Figure 3c Smad7 L4 micrograph, while the pictures in 3e are beautiful, in 3c, "Inhibitor of TGF- β signaling, Smad7, is expressed in barrel cortex in a septa-related pattern" is puzzling: yes, indeed vGlut2-label shows patch TC terminal patterns but barrel cells surround them forming the barrel walls around these patches and a few are located inside the barrel hollows as well. An enlarged view of this picture indicates that the green cells (Smad7+) are not only in the septa but also around the purple patches and in fact, there are a few but sparse green cells inside the purple patches. In this figure it is difficult to differentiate barrel cells from septal cells. In 3e, two of the beautifully green labeled pyramidal cells are inside the barrel TCA patches suggesting that Smad7 may not be an exclusive septal cell marker.

Figure 4: Concerns about the callosal connections (lines 263-284, Figure 4):

Regarding callosal projections of layer 4 cells. Firstly, the old (dated) references (49, 50 from 1997 and 1984) cited for the location of the callosal projection cells are for the rat. One of those studies show complementary patterns of thalamic and callosal projections to barrel centers (in the rat) and septal regions but septal projections come from callosal projection neurons residing in layers II/III and V.

Callosal axons are derived from cortical layer 2/3 and 5 neurons, as the authors acknowledge. There are numerous developmental studies on the callosal projections in the mouse that are better suited (PMID32166131, PMID17942728, PMID: 17942728, PMID: 21129791, PMID: 27307230)

It has been reported that L4 callosal projections in the mouse are transient and are eliminated after the acquisition of local connectivity (PMID31591398). This study, using axonal retrotracing and GFP-targeted visualization of Rorb+ neurons, showed that L4 neurons develop transient interhemispheric axons. Surgical and genetic interventions of sensory circuits further suggested that refinement depends on inputs from sensory thalamic nuclei, and that whether they are connected locally or callosal is dependent on thalamic inputs. The present results need to be interpreted in this context.

Further, it is important to note that previous studies demonstrated that callosal projection patterns are largely dependent by sensory afferent input as removal of peripheral sensory input at birth severely disrupts them (PMID 2304906, 24305168).

Figure 4b. Output connectivity of Smad7OE neurons. Several of the yellow arrows in complementary micrographs are pointing to blank/black spaces instead of double or single labeled neurons. Also related to that figure CTB is often used as an anterograde and retrograde axonal tracer (PMID: 8955961 PMID: 1617444 PMID: 8815303 etc.) It is not clear why the authors got only retrogradely labeled neurons without the anterograde reciprocal axonal projections. In the same figure it also looks like there is a higher density of retrogradely CTB labeled neurons in both layers 3 and 5 and also IUE GFP neurons targeting layer 4 neurons but also labeled cells in layer 5 as well. It would be helpful if the authors discuss these variabilities in their data, rather than attempting to present their case as cut-and-dried.

In Figure 6, while quantification of c-fos expression in control and Smad7OE mice after single whisker texture differentiation appear different (c), the overlay images in (b) (especially the c-fos overlay) do not give that impression.

Minor:

There are some misspells and conceptual errors:

360 line input by the lemniscal/spiny stellate circuit

Figure 6. Functional consequences of spiny stellated-depleted barrel cortex

Lines 200-205 “we hypothesized that certain genes are likely to control upstream events in barrel formation” Barrel cortex is downstream from the whiskers along the pathway lading to the cortex.

Upstream locales would be corticocortical or callosal projections from the barrel cortex or subcortical projections originating from layers 5 and 6 of the barrel cortex.

Lines 493-499 by default “higher order” identity of septal pyramidal cells is too much of an over interpretation.

Reviewer #2 (Remarks to the Author):

This paper addresses the broad issue of cell specification and the role extrinsic signals in cell determination. It focuses on the specification of spiny stellate cells (ssc) in layer 4 of the mouse somatosensory cortex. These neurons differentiate during postnatal life, from pyramidal like neurons that initially have an apical dendrite and long range projections which they lose as maturation proceeds and a barrel-like organization emerges instructed by thalamocortical afferents (TCAs). TCAs are known to be important for the acquisition of the molecular identity of somatosensory cortex and for the dendritic organization of the spiny stellate cells, however the underlying molecular mechanisms remain unclear.

Here the authors address this question taking advantage of a mouse model in which, no thalamic afferents reach the cerebral cortex. They find that in these mutants, ssc do not differentiate and retain their pyramidal-like characteristics, rendering them undistinguishable from neurons present in the nearby barrel septae. A spatial transcriptomic approach then allows them to identify molecular signatures of the ssc vs/ pyramidal cells, which they confirm by an extensive mapping of gene expression, which corresponds to a striking complementary expression of these 2 gene classes in the hollows vw/septae of the barrels. They then identify Smad7 signaling as a major player of this cellular differentiation. They show that Smad7 overexpression in layer 4 neurons renders all neurons pyramidal and septal-like including connectivity and physiological profile, which results in blunting the sharp somatosensory processing from whiskers. Curiously, Smad7 downregulation has no effect.

The work is really remarkable by its experimental qualities, the clarity of the writing and the beauty of the figures. It covers an impressive array of approaches in a logical way and allows to understand how different cell types become committed to different cell fates and acquire different physiological properties when they driven by an external signal which drives the selection, of a specific genetic pattern.

The experiments are convincing and thorough with multiple complementary approaches. My only major concern relates to the possibility of some non specific effects of Smad7 overexpression and the possibility that cell loss could occur in layer 4. Indeed it seems from ext.figure3 that there is a general downregulation of gene expression in layer 4 (ext figure 3 a b) - and the number of transfected neurons are reduced at P7 (ext fig 3-f). Recently Sato et al showed that early deletion of the thalamus reduced

cell numbers in layer 4, which might be the cells that are destined to become the SSC. This could change a bit the interpretation of the results. It would therefore be important to show some cell counts in layer 4 using *rorb* for instance in the OE experiment. Along the same line, one may have wanted to have a bit more illustration of the mouse model – the NEXcre;*Smad7*-Tg mice, where OE may be more controlled. Result in this line only appears in the supplementary material with a histogram (Ex-figure3-G).

Minor remarks

- 1) Figure 1g, labels of genes fall out of the box and pastel color difficult to read. Figure 1e, the colour code is confusing, there seems to be the same amount of dots in the control and KO.
- 2) In the figure legend of Figure 1, keep the wording consistent with the figure, using “hollow gene expression” rather than barrel-related genes. Throughout.
- 3) In extended Figure 1, which is useful, it would be helpful to have the ISH data listed in the same order for all the panels, so that one can more easily compare the tangential and coronal planes. Alphabetical order would be the most straightforward. Also some of the pictures are very dark.
- 3) Figure 4 b: the yellow triangles are a bit too much and prevent really seeing the double labeling. Here again the GFP+ cells seem less numerous in the *Smad7*-OE.
- 4) It is unclear how the *Smad7* gene was selected since it doesn't seem to come up in the spatial transcriptomic analysis. This should be explained.

Reviewer #3 (Remarks to the Author):

Review for Nature Communications

Whisker-related patterns, the “barrels” in the primary somatosensory cortex become visible during the first postnatal week in rodents. The literature provides evidence emphasizing the importance of neurotransmission from either thalamocortical axons or cortical neurons in formation of the barrels. Interestingly, L4 neurons acquire their spiny stellate shape after TCA innervation occurs and depends on neurotransmission at thalamocortical synapses. Despite the recognition of different circuits for barrel hollows vs barrel septa, the underlying molecular mechanism in the establishment of spiny stellate cells has not been revealed. In this manuscript, Young et al. identified large numbers of “barrel hollow” and “barrel septa” marker genes by taking advantages of powerful spatial transcriptomic analysis and *Celsr3* conditional knockout mice lacking TCA innervation of L4. Excitingly, the timing for the expression of hollow- and septa-maker genes coincides with the timing for thalamocortical axon innervation. Examining the morphologies of L4 neurons in the primary somatosensory cortex, the authors found an increased percentage of pyramidal neurons in L4 of *Celsr3* conditional KO S1 cortex than control cortex.

This outcome, together with previous work from Li et al (2013), provides strong evidence to support an instructive role of TCA innervation in guiding the transition of L4 pyramidal neurons to spiny stellate neurons.

The identification of TGF- β family receptors in S1 cortex, coupled with the observation that TGF- β ligands are expressed in the VPM thalamus, prompted the authors to speculate on the importance of TGF- β signaling in mediating TCA guidance during barrel formation and the differentiation of spiny stellate neurons. They took the approach of overexpressing Smad7 (Smad7OE), a TGF- β signaling inhibitor, to prevent L4 neurons from acquiring their spiny stellate morphology. Such Smad7OE in L4 neurons increase septa markers, while hollow markers were reduced. Substantial and solid data acquired from a rather comprehensive battery of approaches were provided to demonstrate that Smad7OE neurons behave like septa/pyramidal neurons. They showed, with heroic efforts, that Smad7OE neurons exhibit several features of L4 pyramidal neurons, including prominent apical dendrite morphology, axonal projections with Smad7OE axons projecting to the contralateral side, receiving wider functional connectivities (demonstrated with both in vivo and ex vivo paradigms). Most importantly, Smad7OE into one hemisphere of S1 L4 disrupted texture discrimination from whiskers on the contralateral side of the face.

Major comments:

Based on Fig 3C ISH, Smad7 seems to express mainly in pyramidal neurons but not L4 spiny stellate neurons based on its septa-enriched expression pattern. However, its expression level does not differ between control and TCA-absent (Celsr3 conditional KO) S1 cortices (Sup Table). Suggesting its abundance is not regulated by TCA innervations. Additionally, they also found no barrel phenotype with Smad7 conditional KO mice. This poses significant concerns on the validity of data derived from Smad7OE for drawing conclusions on the role of endogenous TGF- β signaling in barrel differentiation. Smad7OE is likely to have far-reaching impacts on many genes rather than just the selected genes evaluated in these studies. Loss-of-function studies such as knocking down TGF- β receptors in L4 neurons or knocking down TGF- β ligands in the thalamus should be conducted to demonstrate a necessary role of TGF- β signaling in instructing L4 neurons to acquire spiny stellate morphologies and specify their connectivity.

Additionally, data provide clarification on whether neurotransmission from TCAs is required to activate expression of “barrel hollow” genes in barrels, while down-regulating “barrel septa” genes in barrel hollow regions will significantly increase the significance of the current studies. Li et al. (PMID: 24012009) found that thalamocortical neurotransmission is required for L4 neurons to acquire spiny stellate morphologies. In this manuscript, Young et al. also showed convincing data for a correlation between the timing of the expression of hollow- and septa-maker genes with thalamocortical axon innervation. Additional data demonstrating the up-regulations of barrel hollow marker genes by TCA neurotransmission will be critical in distinguishing the nature of TCA-derived cues.

The authors should discuss why Smad7OE neurons tend to distribute outside of the TCA clusters. According to the author's hypothesis, inhibiting the TGF- β receptor should allow Smad7OE neurons to ignore the influence of TCA-derived signals and thus they should not be displaced outside the barrel hollow.

Minor comments:

1. Unclear why RORc is used for cKO but not ctrl as L4 marker for Fig 1. Panel h images came from both P7/P14. The maturity of L4 is quite different between these two ages. Ages should be labeled for individual images.

2. Methods stated that subjects of both sexes were used for their study. However, there is no consideration of sex when they conducted data analysis and it is also unclear whether similar numbers of both sexes were used in the various experiments. It is important to determine whether there is impact from sex on their behavioral data. I understand the difficulties for some of technically challenging experiments to ensure equal numbers of subjects of each sex. In that case, they should acknowledge that they have not considered sex as a biological variable and future studies will be required to distinguish sex-dependent changes.

3. Using example images they showed that Smad7OE leads to downregulation of hollow-related genes (Btbd3), and upregulation of septa-related genes (Nr1h2 and Sorl1). Quantification of these changes should be provided.

Dear Editor of Nature Communications,

Here we enclose our revised manuscript, "Thalamocortical control of cell-type specificity drives a circuit for processing whisker-related information in mouse barrel cortex", addressing the specific comments and concerns raised by the reviewers. We thank all three reviewers for their time and expertise in order to raise some important issues that we believe has helped to improve the manuscript overall. We have tracked the changes in the revised manuscript and have addressed the points raised by each reviewer below:

- Reviewers' comments in blue
- Responses in black

Reviewer #1

"1. The methods (lines 825-26) note that "Celsr3 conditional knockout (cKO) mice were obtained by crossing female Celsr3^{f/f} mice^{31,54} (RRID:MGI:3579160) with male Celsr3^{f/+};Dlx5/6-Cre^{+/-} mice⁶⁵ (RRID:MGI:3795743)". And the authors note in lines 111-113 that they used Dlx5/6cre;Celsr3^{flox/flox} (Celsr3 conditional knockout) mice which lack TCA innervation of the cortex and as a result, fail to form barrels. There is no further information given about this mouse line which is a concern. Because there are published reports showing that Celsr3 gene is required for interneuron migration in the mouse forebrain (PMID: 19332558); Celsr3 gene is involved in globus pallidus development and connectivity (PMID: 25113559); Celsr3 inactivation in the telencephalon, in the ventral forebrain, or in the cortex have essential roles in both subcerebral projections (including the spinal cord) of cortical neurons and , thalamocortical axons (TCAs) that project through the ventral forebrain (PMID: 19349379) and along with FGz3 regulate reciprocal thalamocortical projections independently of their expression in cortical or thalamic neurons. (PMID: 27170656). Celsr3 is also involved in interneuron migration in the mouse brain (PMID 19332558). Celsr3 ko mice exhibit loss of 50% of glutamatergic synapses in pyramidal neurons of the hippocampus (PMID 28057866). These multitude of Celsr3 functions during brain development deserve some discussion at least."

We thank the reviewer for raising an important point regarding the Celsr3cKO mouse line we have used in this study and agree that a more detailed explanation of this model is needed, given the wide-reaching effects of Celsr3 inactivation. We have provided more rationale for using this Celsr3cKO model and discuss potential off-target effects which should be considered. This has been incorporated into the manuscript and also addressed below.

Celsr3 belongs to the protocadherin family and has important roles in axon outgrowth and guidance, mediating cell adhesion via homophilic interactions. Regarding axon guidance, Celsr3 is required on both axons and guideposts cells along the projection pathway (PMID: 18487195). Indeed, expression of Celsr3 in the ventral telencephalon can regulate reciprocal thalamocortical projections independently of its expression in cortical or thalamic neurons (PMID: 19349379, PMID: 20534841, PMID: 22607003). It is with this cell non-autonomous role which we aimed to test the scenario in which TCA innervation of the cortex is absent, using the Dlx5/6-IRES-cre; Celsr3^{f/f} mouse line (Celsr3cKO^{dlx5/6}). Namely, this cross targets guidepost cells in the vicinity of the internal capsule to disrupt thalamocortical axon pathfinding. This contrasts with a constitutive knockout of Celsr3, or other models of Celsr3 knockout in the telencephalon (Celsr3cKO^{Emx1} or Celsr3cKO^{Foxg1}). Therefore, in the Celsr3cKO^{dlx5/6} mouse line, glutamatergic neurons in the cortex and thalamic projections are likely to have a normal level of Celsr3 expression. Interestingly, barrels still form in Celsr3cKO^{Emx1-cre} brains suggesting a specific role for TCA innervation (PMID: 18487195).

"Celsr3 gene is involved in globus pallidus development and connectivity (PMID: 25113559)"

It is a distinct possibility that development of the globus pallidus (GP) is also affected in Celsr3cKO^{dlx5/6} animals given the majority of its cells are derived from the MGE. How a possible GP deficit would directly impact the early development of barrel cortex is unclear given their indirect connectivity. It is likely, however, that it would further contribute to the pathfinding deficit of thalamocortical axons around the subpallium given its vicinity to the internal capsule/corridor cells (PMID: 27170656, PMID: 22607003). Similar to the Celsr3cKO^{dlx5/6} mouse, targeted knockdown of Celsr3 in the globus pallidus of wildtype mice by *in utero* electroporation had a similar effect on thalamocortical axon guidance. This suggests that the developing GP may assist in the development of the major thalamocortical and corticofugal pathways as guidepost cells in the ventral forebrain (PMID: 25113559).

"Likewise, Dlx5/6 are expressed by developing and mature GABAergic interneurons and play a role in the differentiation of certain GABAergic subclasses (PMID: 31514171, PMID: 35681437)."

We do not believe that levels of Dlx5/6 are impacted in the mouse line we have used since Cre recombinase is expressed from an internal ribosome entry site (IRES; this description of the mouse line is now stated more clearly in the manuscript). Additionally, while there is indeed a study using a constitutive Celsr3 KO animal that suggests a role in interneuron migration, the conditional Celsr3 knockout resulting from a Dlx5/6-IRES-Cre cross is reported to have a similar density and proportions of interneuron subclasses in the cortex (PMID: 20534841). As alluded to in this study, these differences may stem from the fact that altered migration seen in the constitutive knockout occurs in a cortex that is in a more severe atrophic state, thus the cues for interneuron migration may be more compromised. Thus, in

the model used in our study, the knockdown of *Celsr3* in interneurons does not appear to primarily affect cortical migration or maturation. There is a possibility, however, that without proper *Celsr3* expression, interneuron function or integration into cortical circuits may be altered and influence barrel formation events. This should be an important consideration for interpretation of our data since early interneuron populations appear to regulate aspects of cortical maturation (PMID: 26844833, PMID: 26844832).

“There are several important issues that need to be clarified and discussed. Since the *Celsr3* cKO cortex does not get TCAs, one would expect some morphological changes in layer 4 but from the extended Figure 1 it does not seem so, and that the cortical thickness looks remarkably similar. Which is very peculiar if the dense TCA innervation in layer 4 is lacking what makes its thickness equivalent to controls?”

As reported in Zhou et al., at earlier postnatal stages the overall laminar identity and cortical thickness is similar between controls and *Celsr3*cKO^{dlx5/6} mice (PMID: 20534841). Despite the similarities at P7, hallmarks of barrel structures are absent (barrel clustering as seen on Nissl, DAPI, and RORc staining). In the *Celsr3*cKO cortex, progressive differences in thickness are observed with age, such that by P20, significant differences in cell density and thickness can be seen (PMID: 20534841). Similar observations are seen in other TCA mutants at P7 (eg, PMID: 35289744, PMID: 24012009), suggesting that intrinsic growth of the cortex is largely normal up until this point, but failure to receive adequate survival/growth cues from TCAs results in morphological differences in the cortex at later ages.

“Does layer 6 still project back to the VPM even when VPM does not project to the S1 cortex (see references in above paragraph about disrupted reciprocal connections following *Celsr3* mutation) ?”

Due to abnormal axon pathfinding around the internal capsule, the major projections (thalamocortical, corticothalamic, subcerebral) are disrupted in the *Celsr3*cKO^{dlx5/6} genetic cross. Findings from *Celsr3*cKO^{Emx1} brains indicate that subcerebral projections appear to be dispensable for barrel formation (PMID: 18487195). However, this raises the indirect possibility that failure to establish corticothalamic projections may disrupt barrel formation. This should be a consideration, as well as whether additional cortical connections, aside from specific TCA innervation, may contribute to the barrel phenotype seen in *Celsr3*cKO^{dlx5/6} mutants.

“Is this unique to the VPM-S1 circuitry or generalized across thalamic sensory nuclei and their cortical primary sensory targets?”

Although we have not shown it for all sensory thalamic nuclei, all thalamic projections that have a normal trajectory through the internal capsule should be affected.

“Figure 1 indicates that the *Celsr3* cKO VPM projects down to and towards the limbic cortex and the cerebral peduncle, where do they end up? Does this mutation also affect VPM/VPL size?”

In the *Celsr3*cKO^{dlx5/6} mouse, VPM/VPL size is reduced, possibly due to the failure to establish proper reciprocal connections with their targets in the cortex. While the trajectory has been described as running obliquely through the pallidum and amygdala (PMID: 19349379, we have not looked at the final targets of the aberrant TCA projection.

“Figures:

It would be much better if the authors would attend to consistency in scale bars they use in the figures. Even in the same group of micrographs in a sub figure they have one scale bar labeled 100 um and several other differing size scale bars with no labels, are they all 100 um? Several of the photomicrographs in the figures appear as if they are prepared by a miniaturist; difficult to discern what they are supposed to show (even when the pdf is magnified 400x).”
We acknowledge the inconsistency of scale labels and have rectified this to help for better comprehension of the figures.

“Figure 1: Figure 1b is a drawing, it would be better to show actual photomicrographs of sections, for example, Nissl stained to see the absence of the barrels in *Celsr3* cKO cortex.”

We have replaced the schema in Figure 1 with the suggested images taken from Extended Data Fig 1.

“In Figure 1c micrographs are too dark to assess them. It is not clear why *Celsr3* cKO cortex is stained for RORc (also no green staining is visible in the presented micrographs) and the control cortex vGlut2. Presumably due to absence of TCAs in the ko cortex but even so, vGlut2 staining for both genotypes would be a better pair.”

As the reviewer correctly states, RORc was used to ensure that L4 neurons were selected for the analysis in the *Celsr3*cKO, however, we acknowledge that displaying consistent staining between genotypes will be a better comparison. We have now included this in the manuscript (updated Fig. 1c)

“With the exception of top right micrograph from the ko cases the other “pyramidal” cells do not look convincingly pyramidal, in fact they look like cells with retarded dendritic development. In these micrographs the authors should

use the same scale bar, if they are all 100 um, then the left top panel ko sample the cell bodies would be at least twice the size of other pyramidal cells.”

We apologize for the confusion regarding the scale bars – we have double-checked and fixed all scale bars in the manuscript. According to our classification, we regarded those neurons with an apical dendrite as ‘pyramidal’. It is possible that the abnormal dendritic field arises from a lack of extracortical inputs supporting normal growth and development as has been reported in layer 5 of these mice (PMID: 20534841). We have now included a description and comment on this in the manuscript.

“Figure 1h also needs scale bar. I am puzzled by the morphological appearance of the pyramidal neurons in top right frame, they amazingly look like layer 5 pyramids, just underneath layer 4, they have far more apical branches as they approach the pial surface, and also distinct basal dendrites compared to their control partners, which clearly look like star pyramids. Could there be a mix up?”

Similar to other studies (PMID: 31519874 PMID: 31395862), for our classification, we did not distinguish between star pyramidal and pyramidal morphologies. Accordingly, we have changed our description of these neurons to ‘pyramidal-like’. We used nuclear DAPI staining and the above mentioned RORc staining to select labeled neurons in L4 for analysis. Furthermore, an analysis of distances from the pial surface indicated that neurons from both genotypes were sampled from a similar range of depths of the cortex, which was centered within L4. There was no bias of Celsr3cKO positions at the L4/L5 border which could be resulting in L5 neurons being included in our analysis (data not shown).

“Is It possible to show c-fos activity in the Celsr3 cKO cortex following whisker stimulation as a control to ensure that the whisker information isn’t getting there?”

In our hands, it was not possible to detect reliable whisker-evoked c-fos activity in P7 animals. Interestingly, when we attempted this experiment in older animals (P14), we found widespread c-fos expression throughout the cortex of Celsr3 cKO mutants, compared to the whisker-specific activity seen in L4 of controls (data not shown). It is possible that this reflects an overall hyperexcitability of the mutant cortex which can occur without appropriate formative cues from the thalamus (PMID: 31048552).

“Figure 2: vGlut2 staining photos are too dark especially at P2 there should be TCAs in presumptive layer 4 even when I magnify the paper 200 times, I can barely see them. In this figure too there is inconsistent use of scale bars. Figure 2c is difficult to interpret, what is the explanation for DsRed2 labeled layer 3 cells especially at P2?”

In utero electroporation is susceptible to imprecise targeting due to factors involving the timings of pregnancy and transfection. While the majority of labelled neurons are indeed situated in layer 4, a delay in relative timing or persistence of the transfected plasmid in the VZ/SVZ population may sometimes result in labeling of such non-L4 neurons. We attempted to minimize off-targeting using a standardized time for embryonic day 0, and always performing transfections around the same time of day (afternoon). We further utilized L4 markers to ensure that our analyses were restricted to the L4 population. We have now adjusted the scale bars in the figure and have added arrows to indicate the TCA innervation.

“Figure 2c legend reads: “By P4, a clear distinction between spiny stellate (pink) and pyramidal morphologies (blue) could be made, with progressively more spiny stellate neurons found at the edge of barrels (visualized by vGlut2staining)” I can’t see any blue cells with pyramidal morphology, all I can see is a few pink, immature looking pyramidal cells bodies.

In Figure 2c. sparse labeling of layer 4 cells with DsRed2 by IUE at P13.5 has labeled cells in both layer 4 and above and a few blue arrowheads are pointing to black spaces. At P2, “shorter apical dendrites” can also be due to plane of section. “By P4, a clear distinction between spiny stellate (pink) and pyramidal morphologies (blue) could be made” in this picture there is only one pink cell with stellate morphology.”

We thank the reviewer for pointing this out and apologize for the confusion regarding the labelling in Figure 2c. The labeling by pink and blue triangles was intended to indicate the position of the apical dendrite of immature spiny stellate- and pyramidal-shaped neurons. From reading the figure legend, we acknowledge the confusion that this may have caused. This has been rectified in the image and the figure legend text. It is possible that the plane of sectioning could lead to incomplete dendritic morphologies being sampled. Our results, however, are in agreement with a higher resolution time-course study of spiny stellate versus pyramidal morphologies (PMID: 30082783).

“Figure 3: “Overexpression of septa-related gene, Smad7, recapitulates gene expression and morphology changes seen in Celsr3 cKO barrel cortex”. Figure 3b, Tgfb1 why is the VPM in this case notably shrunken? In all sections what is labeled as VPM is actually VPM+VPL. This figure also has inconsistent scale bar use. In d, how were the barrel outlines (dashed lines) determined?”

We again thank the reviewer for highlighting these issues and have adjusted the figure accordingly. A section has been added to the Methods to include a description of the analysis performed here.

Dashed lines in Figure 3d were determined as follows: sections stained with GFP riboprobe were used to firstly assess the region of *in utero* electroporation (IUE). Brains in which the IUE region did not target barrel cortex were not included in downstream analyses. This was determined by overlaying the GFP signal with corresponding serial sections hybridized with Rorb probe, which shows broader, clustered expression in L4 barrel cortex. In some brains, the position of unaltered barrel marker expression in the un-electroporated cortex was used as a proxy. By using the GFP image overlayed with corresponding serial sections that were hybridized with our genes of interest, we outlined a region-of-interest (ROI) that was consistent across sections (although not accounting for slight differences in the anterior-posterior position of serial sections, offset by 28-56µm). A corresponding ROI on the un-electroporated hemisphere was chosen based on the position of the IUE ROI. For analysis of expression, a plot profile was generated and expression intensity was binned along the radial axis (100 bins). Log2FoldChange (ep vs cont side) was calculated.

"This reviewer is a bit puzzled by Figure 3c Smad7 L4 micrograph, while the pictures in 3e are beautiful, in 3c, "Inhibitor of TGF-β signaling, Smad7, is expressed in barrel cortex in a septa-related pattern" is puzzling: yes, indeed vGlut2-label shows patch TC terminal patterns but barrel cells surround them forming the barrel walls around these patches and a few are located inside the barrel hollows as well. An enlarged view of this picture indicates that the green cells (Smad7+) are not only in the septa but also around the purple patches and in fact, there are a few but sparse green cells inside the purple patches. In this figure it is difficult to differentiate barrel cells from septal cells. In 3e, two of the beautifully green labeled pyramidal cells are inside the barrel TCA patches suggesting that Smad7 may not be an exclusive septal cell marker."

This reviewer brings up an excellent point and agree that in the mouse, owing to narrow region of the septa, it is hard to differentiate between barrel and septa cells. That some neurons with pyramidal morphology can be sparsely found inside barrel walls may reflect the immunostaining pattern, however, we concede that without further detailed molecular distinction of hollow and septa neurons it is difficult to claim for certain that Smad7 is an exclusive septa marker, rather appears more enriched in this region.

"Figure 4: Concerns about the callosal connections (lines 263-284, Figure 4):

Regarding callosal projections of layer 4 cells. Firstly, the old (dated) references (49, 50 from 1997 and 1984) cited for the location of the callosal projection cells are for the rat. One of those studies show complementary patterns of thalamic and callosal projections to barrel centers (in the rat) and septal regions but septal projections come from callosal projection neurons residing in layers II/III and V. Callosal axons are derived from cortical layer 2/3 and 5 neurons, as the authors acknowledge. There are numerous developmental studies on the callosal projections in the mouse that are better suited (PMID32166131, PMID17942728, PMID: 17942728, PMID: 21129791, PMID: 27307230)"

While our focus here is on possible changes to the L4 callosal projection, we have edited this section to emphasize that the main callosal projection is indeed from layers 2/3 and 5. Our main reason for citing the dated references in the rat is that they are, to our knowledge, the only references that describe a sparse L4-derived callosal projection in the somatosensory cortex that are located within the L4 septa region of a mature animal (in addition to those derived from L2/3 and L5). We are unaware of any such references looking at this in the mouse. While the references are old, our data suggest a similar situation exists in the mouse. Indeed, the cited study on mouse L4 callosal projections, demonstrated that while this population is transient, there still exist L4 RORb+ neurons in S1 that can be labelled retrogradely by callosal injections of CTB in the adult (eg. PMID31591398: Fig. 1h, 2f, 5c). We also have found in our experiments that these CTB-labelled L4 neurons are RORb+ (data not shown).

"It has been reported that L4 callosal projections in the mouse are transient and are eliminated after the acquisition of local connectivity (PMID31591398). This study, using axonal retrotracing and GFP-targeted visualization of Rorb+ neurons, showed that L4 neurons develop transient interhemispheric axons. Surgical and genetic interventions of sensory circuits further suggested that refinement depends on inputs from sensory thalamic nuclei, and that whether they are connected locally or callosal is dependent on thalamic inputs. The present results need to be interpreted in this context.

Further, it is important to note that previous studies demonstrated that callosal projection patterns are largely dependent by sensory afferent input as removal of peripheral sensory input at birth severely disrupts them (PMID 2304906, 24305168)."

We agree these findings are important in interpreting our data and have tried to highlight this more in the manuscript.

"Figure 4b. Output connectivity of Smad7OE neurons. Several of the yellow arrows in complementary micrographs are pointing to blank/black spaces instead of double or single labeled neurons. Also related to that figure CTB is often used as an anterograde and retrograde axonal tracer (PMID: 8955961 PMID: 1617444 PMID: 8815303 etc.) It is not clear why the authors got only retrogradely labeled neurons without the anterograde reciprocal axonal projections. In the same figure it also looks like there is a higher density of retrogradely CTB labeled neurons in both layers 3 and 5 and also IUE GFP neurons targeting layer 4 neurons but also labeled cells in layer 5 as well. It would be helpful if the authors discuss these variabilities in their data, rather than attempting to present their case as cut-and-dried."

We thank the reviewer for bringing up several important points. We have tried to make the micrographs more clear by showing examples of regions with single and double labeled neurons (updated Fig. 4c).

Although it is not clear in the submitted images, we do indeed see in the same section both retrogradely labeled neurons and anterograde signal (as more diffuse axonal signals). This is more evident at the S1/S2 boundary and consistent with the literature (eg. PMID31591398). We have now provided wider view examples to show this more clearly (updated Fig. 4b). Owing to off-target positioning inherent to *in utero* electroporation (see above), we observed in both controls and Smad7OE brains, transfected neurons in layer 5 whose numbers are variable across brains. We find, however, the effects of Smad7OE appear to be specific to L4. Here we have quantified the density of L2/3 CTB+ neurons as well as the percentage of electroporated L5 neurons which were double-labeled with CTB (updated Fig. 4d-f). Here, no significant differences were found between transfection groups.

"In Figure 6, while quantification of c-fos expression in control and Smad7OE mice after single whisker texture differentiation appear different (c), the overlay images in (b) (especially the c-fos overlay) do not give that impression."

While we have changed the color palette of the figure in order to allow easier perception of these differences, we appreciate that the differences here are subtle thus have tried to clarify these differences in the text. While control animals consistently display very little c-fos expression outside the whisker-spared barrel region, occasional c-fos positive cells are observed in the septa regions, resulting in a subtle, net-like appearance on control overlays. Despite c-fos activity being more widespread outside the barrel in Smad7OE brains, individual differences are more varied (Figure 6d) thus are not as distinctly seen on overlay. We believe, however, that a distinction between the faint septa-like pattern in control overlay and the broader c-fos expression outside the barrel in Smad7OE overlay can be discerned.

Minor:

"There are some misspells and conceptual errors:
360 line input by the lemniscal/spiny stellate circuit"
Corrected to lemniscal

"Figure 6. Functional consequences of spiny stellated-depleted barrel cortex"

Corrected to ...stellate-depleted barrel cortex

"Lines 200-205 "we hypothesized that certain genes are likely to control upstream events in barrel formation" Barrel cortex is downstream from the whiskers along the pathway lading to the cortex. Upstream locales would be corticocortical or callosal projections from the barrel cortex or subcortical projections originating from layers 5 and 6 of the barrel cortex."

Here, we intended this to refer to an early molecular event that is critical to the formation of the barrel cortex, occurring after innervation by thalamocortical input. The current wording may be somewhat ambiguous, therefore have adjusted the text to clarify this. We thank the reviewer for pointing this out.

"Lines 493-499 by default "higher order" identity of septal pyramidal cells is too much of an over interpretation."

We accept that this is too strong of an interpretation and have removed this from the manuscript.

Reviewer #2

“The experiments are convincing and thorough with multiple complementary approaches. My only major concern relates to the possibility of some non specific effects of Smad7 overexpression and the possibility that cell loss could occur in layer 4. Indeed it seems from ext.figure3 that there is a general downregulation of gene expression in layer 4 (ext figure 3 a b) - and the number of transfected neurons are reduced at P7 (ext fig 3-f). Recently Sato et al showed that early deletion of the thalamus reduced cell numbers in layer 4, which might be the cells that are destined to become the ssc. This could change a bit the interpretation of the results.It would therefore be important to show some cell counts in layer 4 using *rorb* for instance in the OE experiment.”

We thank this reviewer for raising an important point. In order to address these issues, we performed cell counts on Control and Smad7OE brains. We did indeed find a significant reduction in L4 density (87% of controls) with Smad7 overexpression. Furthermore, we found a significant reduction in RORb+ L4 neurons (67% of controls). While it is possible that Smad7OE is leading to a selective loss of neurons destined to become spiny stellate neurons, the magnitude of reductions we observe here are not comparable to the majority spiny stellate population seen in controls (being 60-70% of barrel neurons). Furthermore, the degree of changes we find in L4 density are not as severe as changes to gene expression and percentage increase in pyramidal neurons with Smad7OE, suggesting that additional mechanisms other than cell loss or may be contributing to the observed phenotype. Interestingly, our findings are comparable in degree to the reductions caused by thalamic ablations as seen by Sato et al., which may indicate that the abnormal TCA clustering that we observe with Smad7OE or an inability to respond to inputs may contribute to these differences. We have included these data and discussed their implications in the manuscript.

“Along the same line , onne may have wanted to have a bit more illustration of the mouse model – the NEX^{cre};Smad7-Tg mice, where OE may be more controlled. Result in this line only appears in the supplementary material with a histogram (Ex-figure3-G).”

We have now provided additional analyses of the NEX^{cre}; Smad7-Tg mouse line, investigating changes to gene expression and barrel size (updated Extended Data Fig. 6). Similar to our morphological analysis, we found evidence for a dose-dependent effect of Smad7 overexpression in downregulating hollow marker expression and upregulating septa expression. We also noted that while barrels still form in this mice line, the overall size and appearance is altered compared to control animals. Notably, while the barrels of the posteromedial barrel field (PMBF) are smaller but remain distinctly defined in Smad7Tg brains, the more anterior barrels are both consistently smaller and harder to resolve in tangential sections. Importantly, we found no changes in cortical thickness or the size of VPM or V1 in Smad7Tg brains, suggesting a specific effect of Smad7 overexpression in somatosensory cortex. While homozygote Smad7 overexpression appeared to have a greater effect on barrel features, compared to heterozygotes, it was not as dramatic as ectopic Smad7 expression achieved by plasmid transfection via *in utero* electroporation. This is likely due to the more controlled overexpression from the NEX^{cre}; Smad7Tg mouse line, as the reviewer has pointed out.

Minor remarks

“1) Figure 1g, lables of genes fall out of the box and pastel color difficult to read. Figure 1e, the colour code is confusing, there seems to be the same amount of dots in the control and KO.”

2) In the figure legend of Figure 1, keep the wording consistent with the figure, using “hollow gene expression’ rather than barrel -related genes. Throughout.

3) In extended Figure 1, which is useful , it would be helpful to have the ISH data listed in the same order for all the panels, so that one can more easily compare the tangential and coronal planes. Alphabetical order would be the most straightforward. Also some of the pictures are very dark.

3) Figure 4 b: the yellow triangles are a bit too much and prevent really seeing the double labeling. Here again the GFP+ cells seem less numerous in the Smad7-OE.”

We thank the reviewer for pointing out these issues with the figures and text. We have now addressed each of these in the updated manuscript.

“4) It is unclear how the Smad7 gene was selected since it doesn’t seem to come up in the spatial transcriptomic analysis. This should be explained.”

We found that in our initial spatial transcriptomics screen using the *Celsr3cKO* mouse line, read counts for Smad7 were low in both control and cKO. Integration of scRNA-seq data helped to identify Smad7 as a candidate septa-related gene, which we confirmed with *in situ* hybridization of Smad7, showing upregulation in *Celsr3cKO*s (updated Fig. 3d). Combining our list of candidate barrel markers with a list of thalamus-specific genes from our lab observations, we tried to determine whether there was an enrichment for certain ligand-receptor pairs that may have roles in thalamocortical interactions and barrel formation. With this approach, we identified TGF-beta family expression in the thalamocortical pathway. A separate bioinformatics-driven analysis (NicheNet) of potential ligand-receptor interactions in the thalamocortical pathway further confirmed this enrichment (updated Extended Data Fig 4). Therefore, given its pattern of expression and function as an inhibitor of TGF-beta signaling, we tested Smad7 for its role in barrel formation. We have now explained this rationale in more detail within the manuscript.

Reviewer #3

Major comments:

“Based on Fig 3C ISH, Smad7 seems to express mainly in pyramidal neurons but not L4 spiny stellate neurons based on its septa-enriched expression pattern. However, its expression level does not differ between control and TCA-absent (Celsr3 conditional KO) S1 cortices (Sup Table). Suggesting its abundance is not regulated by TCA innervations. Additionally, they also found no barrel phenotype with Smad7 conditional KO mice. This poses significant concerns on the validity of data derived from Smad7OE for drawing conclusions on the role of endogenous TGF- β signaling in barrel differentiation.”

While we did not find a significant difference for Smad7 between Control and Celsr3 conditional KO mice in our spatial transcriptomics data, this is possibly due to overall low Smad7 coverage in our data (low read counts in both controls and cKOs). Integration of our spatial data with scRNA-seq data helped to identify Smad7 as a possible septa candidate (higher expression in septa clusters, updated Extended Data Fig. 2d), while we also found with *in situ* hybridization, an upregulation of Smad7 in Celsr3cKO cortex (updated Fig. 3d).

“Smad7OE is likely to have far-reaching impacts on many genes rather than just the selected genes evaluated in these studies. Loss-of-function studies such as knocking down TGF- β receptors in L4 neurons or knocking down TGF- β ligands in the thalamus should be conducted to demonstrate a necessary role of TGF- β signaling in instructing L4 neurons to acquire spiny stellate morphologies and specify their connectivity.”

We thank the reviewer for this suggestion and have attempted a form of this experiment, namely knocking down Tgfb1 in the cortex using shRNA constructs expressed by *in utero* electroporation (Extended Data Fig. 5e-g). Here, we find that downregulating the receptor reduced the size of barrels, with a smaller area of Btd3 (hollow marker) expression and reciprocal changes to Sorl1 (septa marker). This suggests, together with our findings from the Smad7^{K312E,K316E} mutant, which is unable to interact with the Tgfb1 receptor (PMID: 15148321), a possible role for some form of TGF- β signaling involving the Tgfb1 receptor in barrel development. Given the Tgfb1 shRNA knockdown phenotype is not as severe as Smad7OE and the lack of barrel phenotype in Smad7 conditional KO mice, we hypothesize that the signaling that operates in the setting of barrel formation is non-canonical and/or involving other family members that we also detected in the thalamocortical pathway (Fig. 3a-b).

“Additionally, data provide clarification on whether neurotransmission from TCAs is required to activate expression of “barrel hollow” genes in barrels, while down-regulating “barrel septa” genes in barrel hollow regions will significantly increase the significance of the current studies. Li et al. (PMID: 24012009) found that thalamocortical neurotransmission is required for L4 neurons to acquire spiny stellate morphologies.

In this manuscript, Young et al. also showed convincing data for a correlation between the timing of the expression of hollow- and septa-maker genes with thalamocortical axon innervation. Additional data demonstrating the up-regulations of barrel hollow marker genes by TCA neurotransmission will be critical in distinguishing the nature of TCA-derived cues.”

We thank the reviewer for raising this important point regarding the role of neurotransmission in barrel formation. While an important question, the exact role that activity has in spiny stellate fate and barrel gene expression is currently hard to dissect. In light of findings of the importance of prenatal thalamic spontaneous activity for barrel formation, manipulations to activity starting from these early timepoints may preclude understanding the direct role of neurotransmission at the time of spiny stellate acquisition. We nevertheless tried to investigate roles for TCA neurotransmission using a number of approaches. Unfortunately, we were unable to obtain the mice used by Li et al. (PMID: 24012009) to assess whether barrel gene expression is also altered when TCA glutamatergic neurotransmission is abolished. While not fully equivalent, our findings in Sert-cre;TeNT-Tg mice suggest that TCA neurotransmission is indeed important for regulating barrel gene expression and formation (updated Extended Data Fig 3). Our observations in the Sert-cre; Rim1/2dKO mouse, however, suggest that this activity relationship is perhaps more nuanced. In these mutants, a disruption in calcium-dependent synaptic release (70-90% reduction according to PMID: 28607399) that results in ‘normal’ barrel gene expression and structure is puzzling, however, may either indicate a low threshold for neurotransmission in controlling these events or reflect appropriate cues being provided by largely normal TCA innervation in these animals (PMID: 22553025). Indeed, TCA inputs in Sert-cre;TeNT-Tg mutants do not cluster appropriately in barrel cortex, while Emx1-cre;NR1cKO mutants display reduced levels of both TCA clustering and hollow gene expression (data not shown), thus raise the possibility that activity acting upstream of TCA clustering is a critical step for providing appropriate barrel formation cues. Without additional experiments involving novel tools to more precisely manipulate activity in the thalamocortical pathway, it will be difficult to address these possibilities for this manuscript.

In our hands, early postnatal ectopic activation of the cortex (using chemogenetics or optogenetics) had no effect on barrel gene expression, while chemogenetic activation of thalamic inputs was not feasible using a Sert^{Cre} driver due to off-target expression contributing to post-administration lethality (data not shown).

“The authors should discuss why Smad7OE neurons tend to distribute outside of the TCA clusters. According to the author’s hypothesis, inhibiting the TGF- β receptor should allow Smad7OE neurons to ignore the influence of TCA-derived signals and thus they should not be displaced outside the barrel hollow.”

We agree that the positioning of Smad7OE neurons around TCA clusters is an interesting observation. One possibility is that while these neurons do not respond to TCA-derived cues as would spiny stellate neurons, there may be additional signals that causes them to be repelled by TCA clusters. Alternatively, these neurons have been displaced by remaining spiny stellate neurons that do not express Smad7. This is now discussed in the manuscript.

Minor comments:

“1. Unclear why RORc is used for cKO but not ctrl as L4 marker for Fig 1. Panel h images came from both P7/P14. The maturity of L4 is quite different between these two ages. Ages should be labeled for individual images.”

Whereas we used vGlut2 to determine the positions of L4 barrels in controls, this was not possible in cKOs due to the lack of cortical TCA innervation. Therefore, RORc was used to ensure that L4 neurons were selected for the analysis in the Celsr3cKO, however, we acknowledge that displaying consistent staining between genotypes will be a better comparison and have included this (updated Fig. 1c).

“2. Methods stated that subjects of both sexes were used for their study. However, there is no consideration of sex when they conducted data analysis and it is also unclear whether similar numbers of both sexes were used in the various experiments. It is important to determine whether there is impact from sex on their behavioral data. I understand the difficulties for some of technically challenging experiments to ensure equal numbers of subjects of each sex. In that case, they should acknowledge that they have not considered sex as a biological variable and future studies will be required to distinguish sex-dependent changes.”

For behavioral experiments, we found no significant interaction between sex and genotype and have now stated this along with numbers of each sex in the figure legend (updated Fig. 7). While we are unaware of evidence to suggest sex-specific differences in barrel formation, we have now acknowledged in the Methods that sex was not considered as a biological variable.

“3. Using example images they showed that Smad7OE leads to downregulation of hollow-related genes (Btbd3), and upregulation of septa-related genes (Nrxn2 and Sorl1). Quantification of these changes should be provided.”

We have now included the quantification of these differences in the manuscript (updated Fig. 3f and Methods)

REVIEWERS' COMMENTS:

Reviewer #1 (Remarks to the Author):

The authors have satisfactorily addressed my concerns. Again I congratulate them for this extensive study.

Reviewer #2 (Remarks to the Author):

The authors have responded to all the points raised in my review and added the additional data requested, they also followed all the advice that was intended to improve and clarify their figures.

The discussion is fair and takes into account all the remarks of the reviewer. Overall this is an important paper which besides its main message concerning developmental mechanisms provides useful new resources (such as detailed data on gene expression) that will be very useful for other researchers in the field.

Reviewer #3 (Remarks to the Author):

Young et al have addressed all my concerns with additional experiments and extensive revision. Their comprehensive and novel data provide significant new insights into how thalamocortical axons contribute to cortical circuit wiring.